# D2CACHE: ACCELERATING DIFFUSION-BASED LLMS VIA DUAL ADAPTIVE CACHING

**Yuchu Jiang**[1,2]   **Yue Cai**[1,2]   **Xiangzhong Luo**[1,2*]   **Jiale Fu**[1,2]   **Jiarui Wang**[1,2]
**Chonghan Liu**[3]   **Xu Yang**[1,2]
[1]Key Laboratory of New Generation Artificial Intelligence Technology and Its Interdisciplinary
Applications (Southeast University), Ministry of Education     [2]Southeast University
[3]Qiyuan Tech
kamichanw@seu.edu.cn, xiangzhong.luo@seu.edu.cn

## ABSTRACT

Diffusion-based large language models (dLLMs), despite their promising performance, still suffer from inferior inference efficiency. This is because dLLMs rely on bidirectional attention and cannot directly benefit from the standard key-value (KV) cache as autoregressive models (ARMs) do. To tackle this issue, we introduce *Dual aDaptive Cache* (d²Cache), which is a training-free approximate KV cache framework for accelerating dLLM inference. d²Cache features a two-stage fine-grained selection strategy to identify tokens and adaptively update their KV states at each decoding step, while caching the KV states of the remaining tokens for reuse. Furthermore, d²Cache naturally offers a more reliable decoding alternative, which can enable quasi left-to-right generation and mitigate premature overconfidence in tokens at the end of the sequence. Extensive experimental results on two representative dLLMs (*i.e.*, LLaDA and Dream) demonstrate that d²Cache not only achieves substantial inference speedups, but also yields consistent improvements in generation quality. The code is available at https://github.com/Kamichanw/d2Cache.

## 1 INTRODUCTION

Diffusion models have recently achieved remarkable success in generating continuous data like images (Yang et al., 2023), but text generation—a fundamentally discrete task—has long been dominated by autoregressive models (ARMs) (Touvron et al., 2023; Achiam et al., 2023; Guo et al., 2025). Building on the foundations of ARMs, recent studies have successfully extended diffusion processes to discrete language modeling and further scaled up these models (Nie et al., 2025; Ye et al., 2025; Li et al., 2025c). These diffusion-based large language models (dLLMs) offer several key advantages over ARMs, such as mitigating the "reversal curse" (Berglund et al., 2023) and capturing high-level global semantic patterns (Nagarajan et al., 2025).

Despite their potential, recent dLLMs still face substantial efficiency challenges (Wu et al., 2025). Due to bidirectional attention, dLLMs cannot benefit from the standard key-value (KV) cache as ARMs do. As shown in Figure 1 (a), ARMs leverage causal attention to sequentially generate new tokens and append each new token to the end of the sequence. This autoregressive process naturally enables the reuse of earlier KV states when generating the next token (Li et al., 2024). In contrast, as shown in Figure 1 (b), dLLMs feature an iterative decoding process over a fixed-length sequence, where masked tokens are progressively replaced with decoded tokens. However, under bidirectional attention, updating even a single masked token changes the context seen by all other tokens (Ye et al., 2025; Nie et al., 2025). As a result, the KV states of the entire sequence must be recomputed at each decoding step, making dLLMs inherently incompatible with the standard KV cache.

To address the above efficiency challenges, recent studies (Ma et al., 2025; Wu et al., 2025; Liu et al., 2025; Hu et al., 2025) have explored approximate KV cache to accelerate dLLM inference. These studies build on the following key observation: *for a subset of tokens, their KV states often*

---

*Corresponding Author

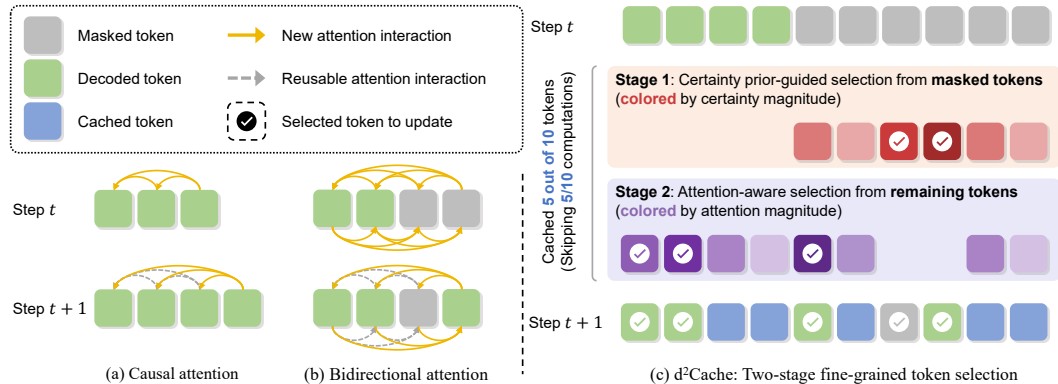

Figure 1: (a) In ARMs, causal attention requires each token to interact only with its preceding tokens. (b) In dLLMs, bidirectional attention requires each token to attend to both its preceding and subsequent tokens, such that any modification in the subsequent tokens necessitates recomputation of the entire sequence. (c) The proposed d²Cache adaptively selects a small subset of tokens in dLLMs and updates their KV states through a two-stage fine-grained process. The KV states of the remaining tokens can be approximately cached for reuse in subsequent decoding step.

*exhibit high similarity across consecutive decoding steps.* This enables to approximately reuse these KV states, which can avoid redundant computations and reduce the overall inference cost. In practice, these studies typically divide the sequence into a static segment, where their KV states can be approximately reused, and a dynamic segment, where their KV states need to be frequently updated within a fixed window of decoding steps. However, these studies are coarse-grained and apply the same strategy to all tokens within both static and dynamic segments. As a result, they either suffer from limited flexibility (Wu et al., 2025) or require complicated tuning (Liu et al., 2025). Moreover, since coarse-grained designs cannot capture the fine-grained token-level dynamics of KV states, they inevitably reuse KV states that should be updated, or update KV states that can be safely reused, thus limiting the achievable acceleration gains.

To address these limitations, we seek to develop an effective fine-grained approximate KV cache strategy, which can adaptively select tokens and update their KV states at each decoding step rather than within a fixed decoding window. To this end, we first perform a fine-grained analysis to investigate the KV state dynamics in dLLMs. Our results show that, for masked tokens, their KV states evolve through three phases: (1) a gradual-change phase during the early decoding steps, (2) a rapid-change phase in the few steps immediately preceding their decoding, and (3) a stable phase after being decoded. Notably, we find that it is sufficient to update the KV states of masked tokens only during the rapid-change phase.

Nonetheless, unlike masked tokens, prompt and decoded tokens exhibit substantially smaller KV state dynamics across consecutive decoding steps. This makes the above phase-based caching strategy less effective and necessitates another caching alternative for prompt and decoded tokens. Inspired by prior KV cache research in ARMs (Feng et al., 2024; Cai et al., 2024), which reveals that attention is unevenly distributed and concentrated on a small subset of tokens—thus allowing to prune the KV states of less important ones—we investigate whether dLLMs exhibit the same attention behavior. Our results confirm that attention in dLLMs is likewise concentrated on a small subset of tokens, especially prompt and decoded tokens. Therefore, similar to KV cache pruning, we can adaptively update the KV states of tokens that receive consistently higher attention, whereas the KV states of the remaining tokens can be safely cached for reuse in subsequent decoding step.

Motivated by the above observations, we propose *Dual aDaptive Cache* (d²Cache), a training-free approximate KV cache framework for accelerating dLLM inference, as shown in Figure 1 (c). Specifically, d²Cache features a two-stage fine-grained selection strategy that identifies tokens and adaptively updates their KV states at each decoding step, while the KV states of the remaining tokens can be cached and reused. In the meantime, d²Cache also naturally delivers a more reliable decoding option, which seamlessly enables quasi left-to-right generation and thus mitigates premature overconfidence in the tokens at the end of the sequence. Extensive experiments on representative dLLMs (*i.e.*, LLaDA (Nie et al., 2025) and Dream (Ye et al., 2025)) demonstrate that d²Cache not

only achieves substantial inference speedups, but also yields consistent improvements in generation quality. Finally, we summarize our main contributions as follows:

- We present a fine-grained analysis on the KV state dynamics in dLLMs, which explicitly reveals a three-phase decoding pattern and uneven attention distribution.

- Building on the above findings, we propose a training-free approximate KV cache framework, namely $d^2$Cache, to accelerate dLLM inference. $d^2$Cache features a two-stage fine-grained selection strategy to identify tokens and adaptively update their KV states at each decoding step, while the KV states of the remaining tokens can be cached for reuse in subsequent decoding step.

- Extensive experiments demonstrate that $d^2$Cache can achieve substantial inference speedups while consistently improving generation quality across various dLLMs and datasets.

## 2 RELATED WORK

**Diffusion-based large language models (dLLMs).** Building on the success of diffusion models in continuous domains, such as image and video generation (Yang et al., 2023; Ho et al., 2022), recent studies have extended diffusion models to discrete language tasks (Sahoo et al., 2024; Shi et al., 2024; Nie et al., 2024; Arriola et al., 2025). Unlike autoregressive models (ARMs) that generate tokens sequentially (Touvron et al., 2023; Achiam et al., 2023; Guo et al., 2025), dLLMs feature an iterative denoising process over masked sequences, which can enable bidirectional context modeling and inherently support parallel decoding (Li et al., 2025c;a;b; Xu et al., 2025). More recently, large-scale dLLMs, such as LLaDA (Nie et al., 2025) and Dream (Ye et al., 2025), have demonstrated competitive performance on reasoning and instruction-following tasks, establishing themselves as a promising alternative to ARMs. Despite their promising performance, their reliance on bidirectional attention necessitates substantial inference overheads, which significantly hinder their practical deployments.

**Approximate KV cache for dLLMs.** Due to bidirectional attention, dLLMs cannot directly benefit from the standard KV cache (Li et al., 2025c) as ARMs do. To address this limitation, recent studies have observed that the KV states in dLLMs remain highly similar across consecutive decoding steps. Building on this observation, several approximate KV caching techniques have recently emerged (Liu et al., 2025; Ma et al., 2025; Wu et al., 2025; Hu et al., 2025). Among them, dLLM-Cache (Liu et al., 2025) partitions the input sequence into two segments—prompt and response—and updates their KV states at different frequencies. dKV-Cache (Ma et al., 2025) introduces a one-step delayed KV caching scheme, in which decoded tokens are stored not at the current decoding step but at the subsequent decoding step. Fast-dLLM (Wu et al., 2025) features block-wise semi-autoregressive decoding and caches all KV states except those in the current decoding block. However, due to the coarse-grained nature, these methods inevitably reuse KV states that should be actively updated or update KV states that can be safely reused, which thus suffer from inferior acceleration gains. A comprehensive comparison between our $d^2$Cache and two concurrent similar works (*i.e.*, dLLM-Cache and Fast-dLLM) is provided in Section B of the Appendix.

## 3 PRELIMINARIES

### 3.1 GENERATION PROCESS OF DLLMS

As shown in Nie et al. (2025), dLLMs feature an iterative denoising paradigm to generate text over $T$ discrete decoding steps, where a fully masked initial sequence is progressively transformed into a fully unmasked final output. Formally, let $\mathcal{V}$ denote the token vocabulary, which includes a special masked token [MASK]. The inference process of dLLMs begins with an initial sequence $y_0$ of length $L$, which is simply constructed by concatenating a prompt segment $p$ with a response segment $r_0$ that consists of $n$ masked tokens. We denote the set of indices corresponding to these masked tokens as $M_0 = \{|p|, |p| + 1, \ldots, |p| + n - 1\}$.

At each decoding step $t \in [0, \ldots, T-1]$, the corresponding sequence $y_t$ is first fed into the given dLLM as input, which produces a probability distribution $p(x_t^i \mid y_t)$ over the vocabulary for each masked position $x_t^i$. Based on this distribution, the most confident token predictions $\hat{X}_t$ and their

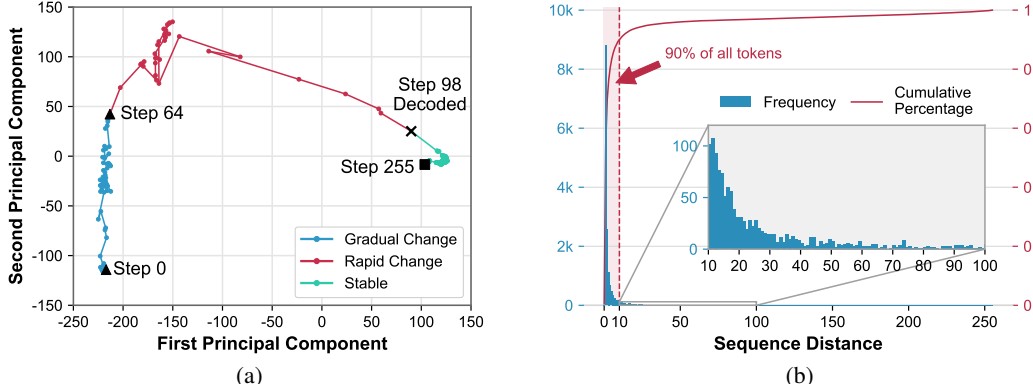

Figure 2: (a) PCA visualization of 77th masked token's trajectory on LLaDA-8B-Instruct with GSM8K ($L$=328, $n$=256, and $T$=256). (b) Sequential distances between token pairs decoded in adjacent steps.

associated confidence scores $S_t$ can be derived as follows:

$$\hat{X}_t = \{\hat{x}_t^i \mid \hat{x}_t^i = \arg\max_{x \in \mathcal{V}} p(x_t^i = x \mid y_t),\ i \in M_t\},$$
$$S_t = \{s_t^i \mid s_t^i = \mathcal{F}\big(p(x_t^i = \hat{x}_t^i \mid y_t)\big),\ i \in M_t\}, \tag{1}$$

where $\mathcal{F}(\cdot)$ is a function that measures the token-level prediction confidence score.

Furthermore, the decoding process employs a scheduling function $\mathcal{G}$ to generate a set of indices $I_t$, which specifies the masked positions in $y_t$ to be replaced with their predicted tokens:

$$I_t = \mathcal{G}(\hat{x}_t, s_t, y_t),\ \text{where } y_{t+1}^i = \begin{cases} \hat{x}_t^i & \text{if } i \in I_t, \\ y_t^i & \text{otherwise.} \end{cases} \tag{2}$$

In practice, the scheduling is typically performed either by randomly sampling a subset of $M_t$ or by choosing those masked positions with the highest confidence scores (Nie et al., 2025). Subsequently, the masked index set for the next decoding step is updated as $M_{t+1} = M_t \setminus I_t$. After $T$ iterations, when the condition $M_T = \emptyset$ holds, the whole generation process is stopped and we get the final sequence $y_T$ with no remaining masked tokens.

## 3.2 KV STATE DYNAMICS AND DECODING ORDER IN DLLMS

Recent studies on approximate KV cache in dLLMs have shown that the KV states of certain tokens exhibit high similarity across adjacent decoding steps (Wu et al., 2025; Liu et al., 2025). Leveraging this redundancy, they first partition the entire sequence into a static segment and a dynamic segment, after which they cache the KV states of tokens in the static segment for reuse. Despite its efficacy, this segment-level partitioning scheme is coarse-grained and totally ignores the fine-grained token-level dynamics. To bridge this gap, we begin with masked tokens and perform experiments on LLaDA-8B-Instruct with GSM8K to explore how their KV states evolve during generation.

**KV state dynamics in dLLMs.** To analyze the dynamics of KV states for masked tokens, we employ principal component analysis (PCA) to project their layer-averaged key states into two dimensions and visualize their trajectories across decoding steps. As shown in Figure 2 (a), the KV states of masked tokens evolve through three phases: (1) a gradual-change phase during the early decoding steps (*i.e.*, steps 0-64), (2) a rapid-change phase in the few steps immediately preceding their decoding (*i.e.*, steps 64-98), and (3) a stable phase after being decoded (*i.e.*, steps 98-255). We find that it is sufficient to update the KV states of masked tokens only during the rapid-change phase, whereas the KV states of masked tokens from the other two phases can be safely cached for reuse. More importantly, this does not degrade the final generation quality, as shown in Figure 5.

**Decoding order in dLLMs.** Building on the above findings, a natural question arises: how can we determine whether a masked token is about to be decoded before its actual decoding—essentially a "*chicken-and-egg*" problem? To shed light on this, we randomly sample 64 examples from GSM8K, in which we analyze the sequential distance between token pairs decoded in adjacent steps. As

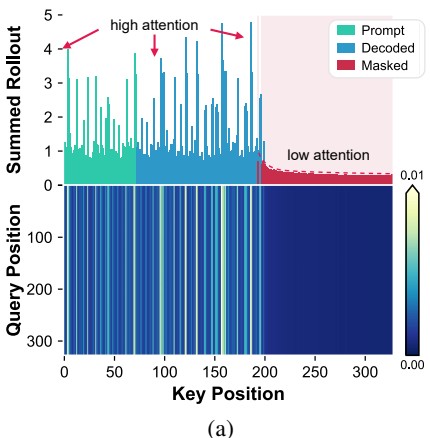 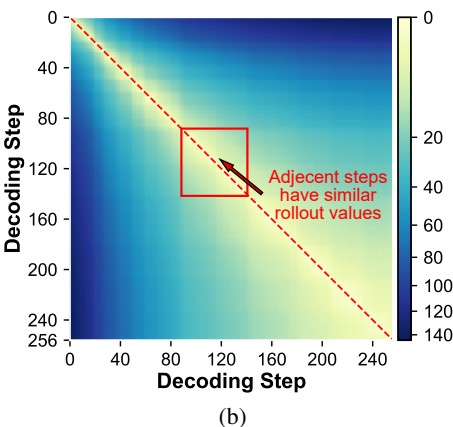

(a)                                     (b)

Figure 3: Attention rollout analysis over sequence, where the example and setting are the same as in Figure 2. (a) Attention rollout visualization at step 126, showing the sum of rollout values over all key positions (*top*) and the pairwise rollout values across different positions (*bottom*). (b) The total absolute differences in rollout values between each two adjacent decoding steps.

shown in Figure 2 (b), LLaDA-8B-Instruct tends to decode the next masked token from positions close to the most recently decoded token, with 90% of tokens falling within a distance of 10. This reveals an interesting decoding pattern: dLLMs tend to decode masked tokens located near previously decoded tokens. Therefore, we can estimate whether a masked token is about to be decoded according to the density of decoded tokens in its local context.

### 3.3 ATTENTION DISTRIBUTIONS IN dLLMS

Prior research on ARMs has observed that attention is not uniformly distributed but instead concentrated on a small subset of salient tokens (Xiao et al., 2023). This observation has served as the foundation for various optimization techniques, which apply differentiated strategies to tokens based on their importance (Feng et al., 2024; Cai et al., 2024; Zhu et al., 2024). This naturally raises the following question: *can the above observation from ARMs generalize to dLLMs*? To answer this question, we conduct experiments on LLaDA-8B-Instruct with GSM8K to analyze the attention distribution.

**Attention salience among tokens.** Inspired by prior attention studies on ARMs, we employ attention rollout (Abnar & Zuidema, 2020) to visualize how attention propagates across tokens. The attention rollout algorithm aggregates cumulative attention by recursively multiplying the attention matrices across layers, yielding a global attribution map that highlights how information propagates from input tokens to the final output. More details about the attention rollout algorithm are provided in Section 4.2. As shown in Figure 3 (a, *bottom*), queries consistently attend to a small subset of key positions in prompt and decoded tokens, revealing that these tokens dominate the attention distribution compared to other tokens. As shown in Figure 3 (a, *top*), masked tokens receive negligible attention, which is substantially lower than that allocated to both prompt and decoded tokens.

**Similarity of attention allocations in adjacent steps.** Building on the above findings, we further calculate the sum of absolute differences in rollout values across all pairs of decoding steps. As shown in Figure 3 (b), the attention allocations across adjacent decoding steps are highly similar. This suggests that the attention allocation of the current decoding step can be used to approximate that of the next decoding step. In light of this, analogous to KV cache optimization techniques in ARMs, KV state updates can thus be restricted to tokens that receive higher attention.

## 4 D$^2$CACHE: DUAL ADAPTIVE CACHE

Motivated by the observations in Section 3, we present *Dual aDaptive Cache* (d$^2$Cache), a training-free approximate KV cache framework for accelerating dLLM inference. To exploit approximate

KV cache in dLLMs, d²Cache seeks to adaptively identify tokens whose KV states should be actively updated at each decoding step, while caching the remaining tokens for reuse in a subsequent decoding step.

**Overview of d²Cache.** We first group tokens in dLLMs into three categories: *prompt tokens*, *masked tokens*, and *decoded tokens*. Based on this categorization, we introduce a two-stage fine-grained token selection strategy. ❶ **Certainty prior-guided selection from masked tokens.** After each forward pass, d²Cache assigns each masked token a score combining prediction confidence and a certainty prior, which represents the density of known tokens (*i.e.*, prompt or decoded ones) in the local context. d²Cache then adaptively selects a subset of masked tokens with higher scores. In light of this, d²Cache naturally enables an alternative decoding scheme guided by the certainty prior rather than prediction confidence alone, yielding higher reliability (see Table 2). ❷ **Attention-aware selection from remaining tokens.** Furthermore, for the remaining tokens (especially prompt and decoded tokens), d²Cache adaptively selects a subset of tokens with higher attention activations. Finally, for the tokens selected in these two stages, d²Cache updates their KV states at each decoding step, while caching the KV states of the remaining tokens for reuse in a subsequent decoding step. An intuitive example of this two-stage token selection is provided in Figure 1 (c).

## 4.1 STAGE 1: CERTAINTY PRIOR-GUIDED SELECTION

As shown in Figure 2 (b), the decoding order in dLLMs is highly localized: 90% of subsequent tokens are decoded within a distance of 10 from the most recently decoded token. Building on this finding, we introduce a *certainty prior*, defined as the density of known tokens (*i.e.*, prompt or decoded tokens) in the local context. In practice, the certainty prior can capture structural certainty, where higher value indicates that the masked token is more likely to be decoded sooner.

Formally, at each decoding step $t \in [0, \ldots, T-1]$, the sequence $y_t$ is fed into the given dLLM to generate predictions $\hat{X}_t$ for the masked tokens $x_t$, together with their corresponding confidence scores $S_t$[1]. With the above in mind, a natural definition of certainty density is the proportion of known tokens (*i.e.*, prompt or decoded tokens) within a fixed local window. However, this definition ignores the effect of relative distance among tokens: intuitively, a known token that is closer to a masked token $x^i$ should impose stronger constraints on $x^i$ than another known token that is farther away. To capture this intuition, we introduce the following position-aware certainty density:

$$D(i) = \sum_{j=0}^{L-1} \phi\left(|i-j|\right) \mathbb{I}_{\{j \notin M\}}, \; s.t. \; \phi\left(|i-j|\right) = \exp\left(-\frac{|i-j|^2}{2\sigma^2}\right), \tag{3}$$

where $i$ denotes the position of the masked token $x^i$ and $j$ denotes the position of each known token in the sequence. In practice, the Gaussian function $\phi(\cdot)$ assigns larger weights to known tokens that are closer to $x^i$ and smoothly diminishes the impact of distant ones, making $D(\cdot)$ a distance-aware aggregation of certainty from all known tokens. The effect of weighting is further controlled by the hyperparameter $\sigma$, which denotes the standard deviation of the Gaussian function $\phi(\cdot)$. A larger $\sigma$ broadens the positional scope considered by $D(i)$, thereby causing the certainty density of different $x^i$ to converge. Finally, we incorporate $D(\cdot)$ into $S$ to measure the certainty prior and select the masked tokens with the top-$k$ calibrated scores, with their indices forming the candidate set $M^*$.

$$M^* = \arg\operatorname{top_k}_{i \in M} \; D(i) \cdot s^i. \tag{4}$$

This formulation ensures that token selection considers both prediction performance and certainty density, which thus can provide a principled foundation for more reliable token selection.

♀ **Certainty prior-guided decoding.** The above certainty prior delivers a novel decoding alternative: masked tokens can be decoded according to their certainty prior rather than their prediction confidence. We demonstrate that the certainty prior-guided decoding can achieve more reliable decoding performance than the default confidence-based decoding, as shown in Table 2. The intuition here is that the certainty prior-guided decoding can preserve a quasi left-to-right decoding order, since masked tokens located closer to known tokens exhibit higher structural and predictive certainty. This quasi left-to-right decoding order effectively mitigates the issue of premature over-confidence in sequence termination during the early decoding steps (Huang et al., 2025).

---

[1]For the simplicity of notation, we omit the subscript $t$ for the current step in the remainder of this paper.

## 4.2 Stage 2: attention-aware selection

In Section 4.1, we present certainty prior-guided selection, which explores masked tokens whose KV states should be updated at each decoding step. In this section, we extend the selection process to the remaining tokens. Notably, we observe that attention rollout (Abnar & Zuidema, 2020)—a widely used attention analysis technique in ARMs—can effectively generalize to dLLMs, particularly for analyzing prompt and decoded tokens, making it well suited for our subsequent token selection.

As described in Abnar & Zuidema (2020), the attention rollout algorithm aggregates cumulative attention by recursively multiplying the attention matrices across layers, yielding a global distribution map that reveals how information propagates from input tokens to the final output. Formally, let $U$ denote the indices of the remaining tokens. At the decoding step $t + 1$, the input of the given dLLM is no longer the full sequence $y_{t+1}$, but instead a subset of it:

$$y_{t+1}^* = \{y_{t+1}^i \mid i \in M^* \cup U\}. \tag{5}$$

To further derive $U$, at each decoding step $t$, we first collect the attention scores $A^{(l)} \in \mathbb{R}^{H \times |y_t^*| \times L}$ from each layer $l \in \{1, \dots, N\}$, where $H$ and $N$ denote the number of attention heads and layers. We then average the resulting attention scores across all heads to obtain $\bar{A}^{(l)}$ and expand $\bar{A}^{(l)}$ into a full-sized attention matrix $E^{(l)} \in \mathbb{R}^{L \times L}$ as follows:

$$E_{i,:}^{(l)} = \begin{cases} \bar{A}_{i,:}^{(l)} & \text{if } i \in M^* \cup U, \\ e_i & \text{otherwise,} \end{cases} \tag{6}$$

where $e_i$ is the one-hot vector with a value of 1 at position $i$. Following Abnar & Zuidema (2020), we further define the per-layer transition matrix $W^{(l)}$ by combining the expanded attention matrix $E^{(l)}$ with the residual connection (*i.e.*, an identity matrix $I$) and applying row-wise normalization:

$$W^{(l)} = \text{normalize}_{\text{row-sum-to-1}}(E^{(l)} + I). \tag{7}$$

The cumulative attention rollout matrix $C$ is then iteratively computed, starting with $C^{(0)} = I$:

$$C^{(l)} = W^{(l)} \cdot C^{(l-1)}. \tag{8}$$

The final rollout matrix $C^{(N)}$ captures the end-to-end influence between all token pairs. To quantify the overall contribution of each token, we further derive an influence score $c_j$ for each token by summing the columns of $C^{(N)}$ as follows:

$$c_j = \sum\nolimits_{i=1}^{L} C_{ij}^{(N)}. \tag{9}$$

Finally, we sort tokens according to their influence scores $c_j$ and directly select the indices of the smallest set whose cumulative probability exceeds the predefined threshold $p$, thus forming $U$.

## 5 Experiments

### 5.1 Experimental setup

**Models, datasets, metrics and hardware.** Following recent conventions (Wu et al., 2025), we evaluate d$^2$Cache on the Base and Instruct variants of two representative dLLMs (*i.e.*, LLaDA-8B (Nie et al., 2025) and Dream-v0-7B (Ye et al., 2025)), which are denoted as LLaDA-Base/Inst and Dream-Base/Inst. Following dLLM-Cache (Liu et al., 2025), we evaluate d$^2$Cache on six benchmarks, including GSM8K (Cobbe et al., 2021), MBPP (Austin et al., 2021), HumanEval (Chen et al., 2021), Math-500 (Lightman et al., 2023), GPQA (Rein et al., 2024), and MMLU-Pro (Wang et al., 2024) to assess performance across diverse reasoning, code generation and general tasks. The performance is reported in terms of task accuracy, which is evaluated using the `lm-eval-harness` framework (Gao et al., 2024). For fair comparisons, we report both inference throughput and latency, where throughput denotes the average number of tokens generated per second and latency denotes the average inference time per sample. All experiments are performed on NVIDIA 3090 24GB GPUs.

**Baselines.** We consider three baselines, including Vanilla and two representative approximate KV cache methods (*i.e.*, dLLM-Cache and Fast-dLLM). For Vanilla, at each decoding step, the masked

Table 1: Comprehensive evaluation results on LLaDA-Inst (Nie et al., 2025) and Dream-Inst (Ye et al., 2025). **Bold** numbers indicate the best results and green texts denote the speedup ratios.

| Dataset | Method | LLaDA-Inst | | | Dream-Inst | | |
|---|---|---|---|---|---|---|---|
| | | Throughput ↑ | Latency(s) ↓ | Score ↑ | Throughput ↑ | Latency(s) ↓ | Score ↑ |
| **GSM8K** *4-shot* Gen. Len. = 256 | Vanilla | 2.77 (1.0×) | 110.26 | 77.6 | 2.62 (1.0×) | 85.94 | 76.7 |
| | + dLLM-Cache | 8.29 (3.0×) | 30.34 | 76.8 | 7.50 (2.9×) | 33.75 | 74.6 |
| | + Fast-dLLM | 9.64 (3.5×) | 26.15 | 77.0 | 10.12 (3.9×) | 24.88 | 77.0 |
| | d²Cache | **11.39** (4.1×) | **22.41** | **79.2** | **12.25** (4.7×) | **21.36** | **78.2** |
| **MBPP** *3-shot* Gen. Len. = 512 | Vanilla | 2.48 (1.0×) | 199.90 | 38.0 | 2.73 (1.0×) | 182.78 | 52.0 |
| | + dLLM-Cache | 6.97 (2.8×) | 71.79 | 38.0 | 7.07 (2.6×) | 71.13 | 52.4 |
| | + Fast-dLLM | 6.80 (2.7×) | 73.27 | 38.4 | 7.29 (2.7×) | 69.47 | 52.0 |
| | d²Cache | **11.42** (4.6×) | **43.86** | **39.4** | **12.47** (4.6×) | **40.32** | **58.0** |
| **HumanEval** *0-shot* Gen. Len. = 512 | Vanilla | 4.99 (1.0×) | 105.76 | 45.1 | 4.39 (1.0×) | 114.86 | 56.7 |
| | + dLLM-Cache | 8.67 (1.7×) | 57.48 | 44.5 | 5.35 (1.2×) | 94.33 | 56.5 |
| | + Fast-dLLM | 7.90 (1.6×) | 63.12 | 43.9 | 7.89 (1.8×) | 63.84 | 56.1 |
| | d²Cache | **14.00** (2.8×) | **35.44** | **48.2** | **14.06** (3.2×) | **36.61** | **61.6** |
| **Math-500** *4-shot* Gen. Len. = 256 | Vanilla | 3.08 (1.0×) | 82.51 | **38.4** | 3.51 (1.0×) | 71.05 | **45.2** |
| | + dLLM-Cache | 6.71 (2.2×) | 37.84 | 38.2 | 7.19 (2.0×) | 35.36 | 44.2 |
| | + Fast-dLLM | 10.61 (3.4×) | 23.79 | 38.0 | 10.72 (3.1×) | 23.52 | 44.4 |
| | d²Cache | **12.02** (3.9×) | **20.19** | 38.0 | **13.80** (3.9×) | **18.80** | 44.6 |
| **GPQA** *0-shot* Gen. Len. = 256 | Vanilla | 6.14 (1.0×) | 43.34 | 25.2 | 6.43 (1.0×) | 41.14 | 30.1 |
| | + dLLM-Cache | 11.51 (1.9×) | 22.33 | 27.2 | 10.91 (1.7×) | 23.62 | 31.0 |
| | + Fast-dLLM | 12.41 (2.0×) | 20.66 | 25.7 | 11.75 (1.8×) | 21.79 | **34.6** |
| | d²Cache | **15.04** (2.4×) | **17.08** | 28.4 | **14.65** (2.3×) | **17.52** | 31.5 |
| **MMLU-Pro** *5-shot* Gen. Len. = 256 | Vanilla | 1.76 (1.0×) | 152.62 | 37.5 | 2.15 (1.0×) | 126.31 | **47.9** |
| | + dLLM-Cache | 6.79 (3.9×) | 38.29 | **38.1** | 7.82 (3.6×) | 34.09 | 46.5 |
| | + Fast-dLLM | 8.91 (5.1×) | 29.00 | 37.1 | 9.74 (4.5×) | 27.69 | 45.9 |
| | d²Cache | **9.59** (5.4×) | **27.60** | 33.1 | **10.12** (4.7×) | **25.77** | 46.8 |
| **AVG** | Vanilla | 3.54 (1.0×) | 115.73 | 43.6 | 3.64 (1.0×) | 103.68 | 51.4 |
| | + dLLM-Cache | 8.16 (2.3×) | 43.01 | 43.8 | 7.64 (2.1×) | 48.71 | 50.9 |
| | + Fast-dLLM | 9.38 (2.7×) | 39.33 | 43.4 | 9.59 (2.6×) | 38.53 | 51.7 |
| | d²Cache | **12.24** (3.5×) | **27.76** | **44.4** | **12.89** (3.5×) | **26.73** | **53.4** |

Table 2: Comparisons of different decoding schemes under the default NAR setting, where `Conf` denotes the confidence-based decoding and `CP` denotes our certainty prior-guided decoding.

| Method | LLaDA-Inst | | | | | Dream-Inst | | | | |
|---|---|---|---|---|---|---|---|---|---|---|
| | GSM8K | MBPP | HumanEval | Math-500 | AVG | GSM8K | MBPP | HumanEval | Math-500 | AVG |
| Semi-AR (Vanilla) | 77.6 | 38.0 | 45.1 | 38.4 | 49.8 | 76.7 | 52.0 | 56.7 | **45.2** | 57.6 |
| NAR w/ `Conf` | 57.5 | 18.2 | 42.1 | 26.4 | 36.0 | 51.6 | 34.2 | 26.8 | 3.2 | 29.0 |
| NAR w/ Only CP | 79.0 | 38.8 | 44.5 | **39.0** | 50.3 | 78.1 | **59.2** | 54.3 | 43.6 | 58.8 |
| Semi-AR w/ d²Cache | 76.9 | **40.2** | 46.6 | 38.6 | 50.6 | 76.0 | 53.8 | 56.7 | 42.0 | 57.1 |
| NAR w/ d²Cache | **79.2** | 39.4 | **48.2** | 38.0 | **51.2** | **78.2** | 58.0 | **61.6** | 44.6 | **60.6** |

position with the highest confidence is replaced with its predicted token. For dLLM-Cache and Fast-dLLM, we employ their default configurations as reported in Liu et al. (2025); Wu et al. (2025). For Instruct variants, all baselines adopt block-wise semi-autoregressive decoding (semi-AR) with a block size of 32, whereas the Base variants are evaluated in fully non-autogressive (NAR) manner. More details are provided in Section C of the Appendix.

**Implementation details.** Unless otherwise specified, the standard deviation $\sigma$ of the Gaussian function is set to 10.0, the number of masked tokens selected per step is fixed at 32, the cumulative probability threshold $p$ is set to 0.1, and the decoding is performed under the certainty prior.

## 5.2 MAIN RESULTS

The evaluation results on LLaDA-Inst and Dream-Inst are summarized in Table 1. Notably, we observe that d²Cache achieves the best overall performance on average across all benchmarks, which delivers the highest throughput, the lowest latency, and the best score, consistently outperforming all baselines. Across all models and datasets, our d²Cache obtains an average 3.5× speedup over Vanilla. Taking Dream-Inst on GSM8K as an example, our d²Cache improves the inference through-

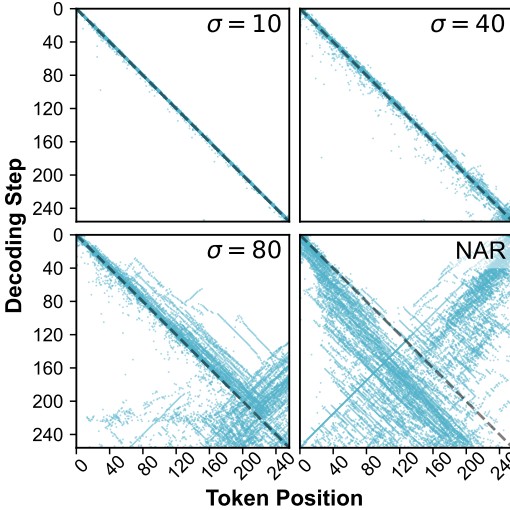

Figure 4: Visualization of the decoding order using certainty prior with different $\sigma$ and NAR decoding. Each dot at $(i, t)$ indicates that the token at position $i$ is decoded at step $t$.

Figure 5: Comparisons of different update strategies, including updating tokens only during the rapid-change phase (Phase 2 Only) and updating tokens during both the gradual-change and rapid-change phases (Phases 1 & 2).

put from 2.62 to 12.25 tokens per second, leading to 4.7× inference speedup. More importantly, these substantial inference speedups are achieved without sacrificing accuracy, as the attainable score on average across six datasets remains comparable to or better than Vanilla. Furthermore, compared to dLLM-Cache and Fast-dLLM, our d²Cache can also deliver better performance in terms of both inference efficiency and accuracy. For example, compared to Fast-dLLM, our d²Cache yields 1.3× inference speedup on Dream-Inst, while maintaining +1.7% accuracy on average across six datasets. These results clearly demonstrate the efficacy of d²Cache, which benefits from its two-stage fine-grained selection strategy.

## 5.3 ABLATIONS AND ANALYSIS

**Certainty prior-guided decoding vs. confidence-based decoding.** As discussed in Section 4.1, d²Cache naturally delivers an alternative decoding scheme: masked tokens can be decoded according to their certainty prior rather than their prediction confidence. To evaluate its efficacy, we further compare our certainty prior-guided decoding with the standard confidence-based decoding under the default NAR setting. As shown in Table 2, our certainty prior-guided decoding delivers more reliable performance than the confidence-based decoding under the default NAR setting. We also observe that certainty prior–guided decoding and semi-AR decoding achieve comparable performance, because both approaches constrain the model to decode in a quasi left-to-right manner. Although they share a similar intuition, only the combination of certainty prior–guided decoding and d²Cache delivers the best performance among all evaluated configurations.

**Effect of $\sigma$ on decoding order.** We visualize the decoding step for each masked position using LLaDA-Inst on 64 randomly sampled examples from GSM8K. As shown in Figure 4, we compare NAR decoding with our certainty prior-guided decoding, where the hyperparameter $\sigma$ (see Equation (3)) is set to 10, 40, and 80. We find that NAR decoding exhibits a distinctive "U-shaped" trajectory: tokens at both sequence boundaries are first generated, which then converge towards the center (Huang et al., 2025). At the first glance, this behavior seems inconsistent with our earlier observation that dLLMs tend to prioritize decoding masked tokens adjacent to known tokens (*i.e.*, prompt or decoding tokens). This discrepancy, however, stems from the supervised fine-tuning (SFT) of LLaDA-Inst, where the excessive number of [EOS] tokens in the training data biases the model towards producing an unnatural number of [EOS] tokens during inference (Nie et al., 2025). In contrast, our certainty prior-guided decoding yields a more natural and controllable left-to-right generation order, where a smaller $\sigma$ makes the generation closer to autoregressive decoding.

**Computational redundancy during the gradual-change phase.** As discussed in Section 3.2, the KV states of masked tokens evolve through three phases: *gradual-change*, *rapid-change*, and *stable*. It is thus natural to update the KV states of masked tokens during both the gradual-change and rapid-

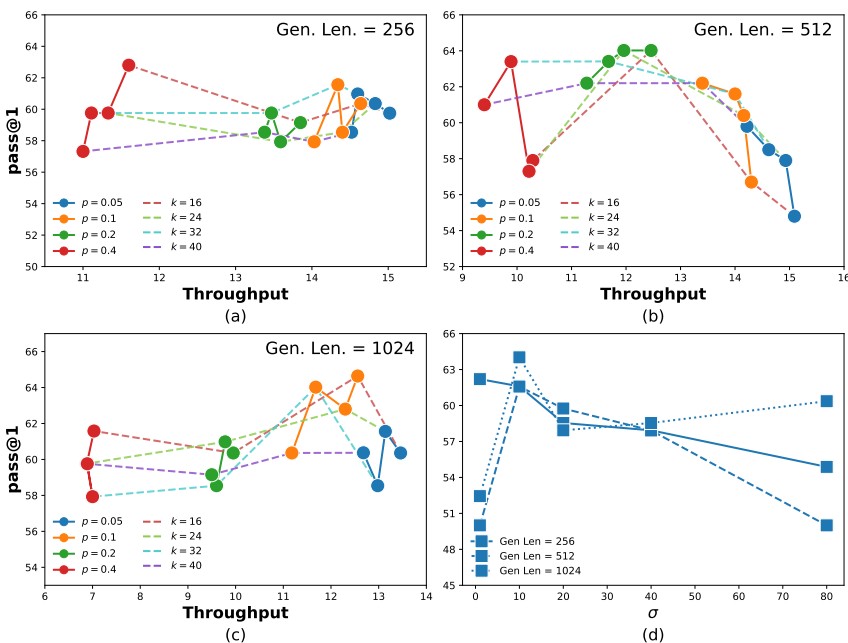

Figure 6: Hyperparameter sensitivity analysis of $p$, $k$, and $\sigma$ on Dream-Inst and HumanEval.

change phases, while caching them for reuse during the stable phase. However, our analysis shows that it is sufficient to update the KV states of masked tokens only during the rapid-change phase. To shed light on this, we conduct an ablation on Dream-Inst, in which we compare the full-update strategy (updating tokens during both the gradual-change and rapid-change phases) with our default selective-update strategy (updating tokens only during the rapid-change phase). As shown in Figure 5, our default selective-update strategy (*i.e.*, Phase 2 Only) delivers higher inference throughput than the full-update strategy (*i.e.*, Phases 1 & 2), while maintaining a comparable or even better score. This finding reveals a counterintuitive property of dLLMs: *increased computation does not necessarily translate into improved performance*. Instead, selectively updating only the most critical tokens can reduce computational redundancy and, in some cases, even yield better performance.

**Hyperparameter sensitivity analysis.** To determine the optimal hyperparameters, we conduct systematic experiments on Dream-Inst and HumanEval with generation lengths of 256, 512 and 1024. As shown in Figure 6 (a-c), the number of masked tokens updated per step is the dominant factor: performance improves as $k$ increases but saturates—and may slightly decline—beyond $k = 32$, indicating that $k = 32$ offers the most stable gains across settings of $p$ and sequence lengths. The cumulative probability threshold $p$, which regulates the retained probability mass and thus affects throughput, does not monotonically improve performance with larger values. We additionally examine the Gaussian standard deviation $\sigma$ in Equation (3), which governs the locality of certainty-prior selection.

# 6  CONCLUSION

In this paper, we propose *Dual aDaptive Cache* (d$^2$Cache), a training-free approximate KV cache framework for accelerating dLLM inference. Through a fine-grained analysis of KV state dynamics, we uncover two key insights behind dLLMs: (1) the KV states of masked tokens exhibit substantial changes only in the few steps immediately preceding their decoding, indicating that their KV states can be reused beyond this phase; and (2) attention distributions are highly skewed towards a small subset of prompt and decoded tokens, indicating that the KV states of low-attention tokens can be reused. Building on these insights, d$^2$Cache introduces a two-stage fine-grained selection strategy that adaptively identifies tokens and updates their KV states at each decoding step, whereas the KV states of the remaining tokens can be safely cached for reuse in subsequent decoding step, thus substantially reducing redundant computations and improving inference efficiency. Extensive experiments on representative dLLMs (*i.e.*, LLaDA and Dream) demonstrate that d$^2$Cache achieves substantial inference speedups, while also yielding consistent improvements in generation quality.

ACKNOWLEDGMENT

This work is supported by Jiangsu Province Carbon Peak Carbon Neutrality Science and Technology Innovation Special Fund Project (Grant No. BT2025029); National Natural Science Foundation of China(62576091); the Southeast University Big Data Computing Center; Southeast University Kunpeng & AscendCenter of Cultivation.

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

# A    THE USE OF LARGE LANGUAGE MODELS

In this work, we employ large language models (LLMs) as general-purpose auxiliary tools, which are mainly used in the following two scenarios:

- **Writing and editing**: LLMs assist in revising the manuscript by enhancing its clarity, grammar, and stylistic consistency.

- **Code generation**: LLMs assist in programming tasks, including debugging and generating illustrative code snippets.

The authors are fully responsible for the entire content of this paper, including sections in which LLMs provide writing assistance. We note that LLMs are not involved in research ideation, experimental design, or data analysis, and therefore do not meet the criteria for authorship.

# B    RELATIONSHIPS WITH CONCURRENT WORKS

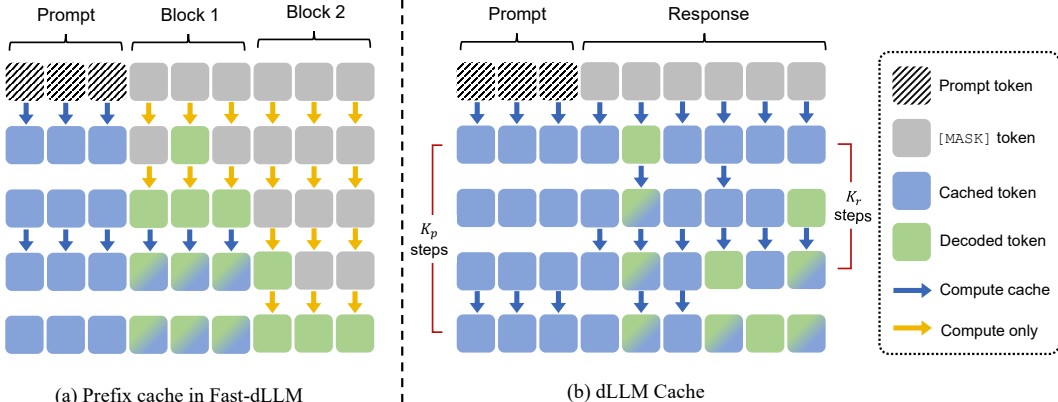

(a) Prefix cache in Fast-dLLM      (b) dLLM Cache

Figure 7: Illustration of existing approximate KV cache works. (a) In Fast-dLLM, the tokens of the current block and all subsequent blocks are recomputed. Once a block has been fully decoded, the KV cache at all positions is refreshed. (b) In dLLM-Cache, the prompt and response update their corresponding segment cache at intervals of $K_p$ and $K_r$ steps, respectively. During steps when the response is not updated, a subset of response tokens is still updated in each layer.

We note two concurrent works on approximate KV cache for dLLMs, including dLLM-Cache (Liu et al., 2025) and Fast-dLLM (Wu et al., 2025). While both share the same motivations of accelerating dLLM inference through approximate KV cache, our $d^2$Cache is fundamentally different.

First and foremost, as shown in Figure 7, dLLM-Cache and Fast-dLLM both operate **at the coarse-grained segment level**, which partition the input sequence into multiple segments and apply different KV state updates to each segment. For instance, dLLM-Cache divides the input sequence into two segments—prompt and response—and updates their KV states at different frequencies. Similarly, Fast-dLLM relies on block-wise semi-autoregressive decoding, which divides the input sequence into multiple blocks (or segments) and sequentially generates these blocks from left to right with tailored KV state updates to each block. Nonetheless, due to the coarse-grained nature, dLLM-Cache and Fast-dLLM inevitably reuse KV states that should be updated or update KV states that can be reused, thus limiting the achievable inference gains.

In contrast, our $d^2$Cache operates **at the fine-grained token level**, which adaptively identifies tokens whose KV states should be updated at each decoding step, while caching the KV states of the remaining tokens for reuse in subsequent decoding step. Thanks to the fine-grained token selection, our $d^2$Cache achieves significant inference speedups while maintaining strong generation quality across different tasks, compared to both dLLM-Cache and Fast-dLLM.

## C    BASELINE HYPERPARAMETERS

In this section, we provide more details about the hyperparameter configurations for the baseline methods (*i.e.*, Fast-dLLM (Wu et al., 2025) and dLLM-Cache (Liu et al., 2025)) across different models and datasets. For Fast-dLLM, we closely follow common practices in prior work and set the block size to 32 for all models (Wu et al., 2025). For dLLM-Cache, we consider its key hyperparameters $K_p$ and $K_r$, where $K_p$ denotes the prompt refresh interval and $K_r$ denotes the response refresh interval. To ensure fair comparisons, we employ the default configurations as reported in Liu et al. (2025), which are also summarized in Table 3.

Table 3: Configurations of dLLM-Cache. $K_p$ and $K_r$ are the refresh interval of prompt and response.

| Dataset | Model | $K_p$ | $K_r$ |
|---|---|---|---|
| GSM8K | LLaDA-8B-Base | 25 | 5 |
| | LLaDA-8B-Instruct | 50 | 7 |
| | Dream-v0-7B-Base | 100 | 8 |
| | Dream-v0-7B-Instruct | 25 | 2 |
| HumanEval | LLaDA-8B-Base | 50 | 5 |
| | LLaDA-8B-Instruct | 25 | 5 |
| | Dream-v0-7B-Base | 5 | 1 |
| | Dream-v0-7B-Instruct | 50 | 1 |
| Math-500 | LLaDA-8B-Base | 50 | 8 |
| | LLaDA-8B-Instruct | 50 | 1 |
| | Dream-v0-7B-Base | 100 | 4 |
| | Dream-v0-7B-Instruct | 50 | 1 |
| MBPP | LLaDA-8B-Base | 25 | 4 |
| | LLaDA-8B-Instruct | 100 | 5 |
| | Dream-v0-7B-Base | 25 | 8 |
| | Dream-v0-7B-Instruct | 10 | 8 |
| GPQA | LLaDA-8B-Instruct | 50 | 6 |
| | Dream-v0-7B-Base | 100 | 8 |
| | Dream-v0-7B-Instruct | 10 | 8 |
| MMLU-Pro | LLaDA-8B-Base | 100 | 6 |
| | LLaDA-8B-Instruct | 50 | 3 |
| | Dream-v0-7B-Base | 25 | 2 |
| | Dream-v0-7B-Instruct | 5 | 1 |

## D    DISCUSSIONS

### D.1    MEMORY OVERHEAD OF CACHING

We conduct a thorough analysis and profiling of the memory overhead of caching. Note that the KV cache used by dLLMs consumes the same amount of memory as that required by an autoregressive LLM (ARM) of the same scale. Specifically, for sequence length $L$, number of layers $N$, and hidden dimension $d$, an ARM or a dLLM stores $2 \times L \times N \times d$ floating-point values for the KV cache. d$^2$Cache additionally stores an attention-rollout matrix of size $L \times L$, which is typically negligible. We report the peak memory usage on Dream Inst for a generation length of 1024 across four datasets. As shown in the Table 7, for example on GSM8K, where the average prompt length is approximately 800—resulting in a sequence length of roughly 1.8k—the additional memory consumption of d$^2$Cache is nearly identical to that of Fast-dLLM (Wu et al., 2025).

### D.2    DECODING ORDER OF DLLMS

In Section 4.1, we proposed certainty-prior–guided decoding, which forces the model to generate in a quasi–left-to-right order. A natural question arises: if dLLMs behave more like autoregressive

Table 4: Comparisons of using only the first 5 layers to compute attention rollout (Rollout-5) and using all layers to compute attention rollout (Full-rollout) on Dream-Inst. **Bold** numbers indicate the best scores, and green texts denote the speedup ratios relative to the Vanilla method.

| Dataset | Method | Throughput ↑ | Score ↑ |
|---|---|---|---|
| **GSM8K** | Vanilla | 2.62(1.0×) | 76.7 |
| *4-shot* | Rollout-5 | 12.61(4.8×) | 71.8 |
| Gen. Len. = 256 | Full-rollout (Ours) | 12.25(4.7×) | **78.2** |
| **MBPP** | Vanilla | 2.73(1.0×) | 52.0 |
| *3-shot* | Rollout-5 | 13.10(4.8×) | 57.2 |
| Gen. Len. = 512 | Full-rollout (Ours) | 12.47(4.6×) | **58.0** |
| **HumanEval** | Vanilla | 4.39(1.0×) | 56.7 |
| *0-shot* | Rollout-5 | 14.20(3.2×) | **62.2** |
| Gen. Len. = 512 | Full-rollout (Ours) | 14.06(3.2×) | 61.6 |
| **Math-500** | Vanilla | 3.51(1.0×) | **45.2** |
| *4-shot* | Rollout-5 | 13.99(4.0×) | 40.2 |
| Gen. Len. = 256 | Full-rollout (Ours) | 13.80(3.9×) | 44.6 |

models (ARMs), does this violate the original intention of enabling parallel, any-order generation? Here, we argue that the answer is not simply "no".

Unlike ARMs, which can generate tokens only at the immediately adjacent next position, dLLMs produce predictions over the entire sequence, which is the source of their any-order generation capability. However, high-quality predictions are not available at all positions. Thus, selecting which tokens to decode—that is, determining the decoding order—is crucial for generation quality. As shown in Nie et al. (2025), LLaDA-Instruct tends to become prematurely overconfident in EOS tokens near the end of the sequence, and therefore proposes block-wise semi-autoregressive decoding (semi-AR), which constrains the model to decode from left to right at the block level while generating in parallel within each block. Compared with fully non-autoregressive decoding, block-wise semi-AR preserves the model's sequential reasoning ability to a large extent, as shown in Table 2.

In our paper, experiments in Section 3.2 show that a dLLM consistently prefers to decode tokens close to known positions. This observation explains why block-wise semi-AR is effective: enforcing quasi-left-to-right generation ensures that each token is decoded only when the contextual information is sufficiently rich. Our certainty-prior decoding shares the same intuition, but provides a more conceptual formulation.

Although dLLMs need to decode in a quasi–left-to-right order to maintain sequential reasoning ability, they still retain substantially greater flexibility during generation. For example, when the model encounters a position where all next-token candidates have low confidence, an AR model must commit to one choice. In contrast, a dLLM can decode further positions and delay the decision until the extended context provides adequate evidence, thereby exploiting its non-AR modeling capacity. A concrete example is pronoun resolution in ambiguous contexts. Suppose the prompt is: "Alice thanked Mary because ＿＿＿ had helped with the project".

At the blank position, an AR model must immediately choose between "she" and "Alice", even though the correct antecedent remains unclear without additional context. A dLLM, however, can tentatively consider both possibilities, continue decoding subsequent positions, and use the extended context to determine whether the sentence is likely to continue as "... she had provided key data", or "... Alice needed assistance", before committing to the final token.

Moreover, even under quasi-AR decoding, each masked token still attends to the entire context (unlike ARMs, where tokens can only attend to previous positions), so the original advantages of dLLMs, such as bidirectional modeling and parallel decoding, remain preserved.

Table 5: Comprehensive evaluation results on LLaDA-Base (Nie et al., 2025) and Dream-Base (Ye et al., 2025). **Bold** numbers indicate the best results and green texts denote the speedup ratios.

| Dataset | Method | LLaDA-Base | | | Dream-Base | | |
|---|---|---|---|---|---|---|---|
| | | Throughput ↑ | Latency(s) ↓ | Score ↑ | Throughput ↑ | Latency(s) ↓ | Score ↑ |
| **GSM8K** *4-shot* Gen. Len. = 256 | Vanilla | 2.31 (1.0×) | 112.39 | 70.4 | 2.67 (1.0×) | 96.29 | 71.7 |
| | + dLLM-Cache | 7.72 (3.3×) | 33.30 | 69.3 | 9.28 (3.5×) | 27.88 | 64.7 |
| | + Fast-dLLM | 9.94 (4.3×) | 25.82 | 70.3 | 10.07 (3.8×) | 25.44 | 71.4 |
| | d²Cache | **11.37** (4.9×) | **22.57** | **72.1** | **12.37** (4.6×) | **21.74** | **73.5** |
| **MBPP** *3-shot* Gen. Len. = 512 | Vanilla | 2.52 (1.0×) | 195.59 | **39.2** | 2.81 (1.0×) | 177.14 | 51.4 |
| | + dLLM-Cache | 6.52 (2.6×) | 77.20 | 38.6 | 7.73 (2.8×) | 64.75 | 49.8 |
| | + Fast-dLLM | 7.00 (2.8×) | 73.16 | 37.4 | 7.07 (2.5×) | 72.40 | 52.6 |
| | d²Cache | **8.62** (3.4×) | **43.41** | 38.0 | **12.67** (4.5×) | **40.10** | **53.6** |
| **HumanEval** *0-shot* Gen. Len. = 512 | Vanilla | 5.02 (1.0×) | 100.54 | 32.3 | 5.45 (1.0×) | 92.11 | 51.2 |
| | + dLLM-Cache | 9.04 (1.8×) | 55.60 | 31.7 | 5.47 (1.0×) | 91.72 | 51.8 |
| | + Fast-dLLM | 8.13 (1.6×) | 62.96 | 31.7 | 7.81 (1.4×) | 88.02 | 53.7 |
| | d²Cache | **14.36** (2.9×) | **35.60** | **33.5** | **14.36** (2.6×) | **37.18** | **61.0** |
| **Math-500** *4-shot* Gen. Len. = 256 | Vanilla | 3.14 (1.0×) | 80.44 | **32.2** | 3.55 (1.0×) | 71.54 | 39.0 |
| | + dLLM-Cache | 9.83 (3.1×) | 25.94 | 29.6 | 9.70 (2.7×) | 26.08 | 35.2 |
| | + Fast-dLLM | **10.93** (3.5×) | 23.41 | 27.0 | 10.69 (3.0×) | 23.94 | 39.4 |
| | d²Cache | 10.80 (3.4×) | **20.13** | 30.4 | **13.86** (3.9×) | **18.63** | **39.6** |
| **GPQA** *0-shot* Gen. Len. = 256 | Vanilla | 6.27 (1.0×) | 42.35 | 30.4 | 6.54 (1.0×) | 40.55 | 32.8 |
| | + dLLM-Cache | 11.32 (1.8×) | 22.69 | **31.0** | 11.12 (1.7×) | 23.10 | **34.6** |
| | + Fast-dLLM | 12.67 (2.0×) | 20.24 | **31.0** | 11.92 (1.8×) | 21.45 | 31.5 |
| | d²Cache | **15.32** (2.4×) | **16.77** | 30.8 | **13.02** (2.0×) | 18.64 | 32.6 |
| **MMLU-Pro** *5-shot* Gen. Len. = 256 | Vanilla | 1.53 (1.0×) | 143.45 | 38.1 | 2.13 (1.0×) | 127.08 | **46.1** |
| | + dLLM-Cache | 6.86 (4.5×) | 37.96 | 37.4 | 7.45 (3.5×) | 34.81 | 44.6 |
| | + Fast-dLLM | 8.96 (5.9×) | 28.83 | **40.0** | 9.42 (4.4×) | 27.31 | 45.9 |
| | d²Cache | **9.58** (6.3×) | **27.60** | 39.1 | **9.71** (4.6×) | **26.73** | 44.4 |
| **AVG** | Vanilla | 3.47 (1.0×) | 112.46 | 40.4 | 3.86 (1.0×) | 100.79 | 48.7 |
| | + dLLM-Cache | 8.55 (2.5×) | 42.12 | 39.6 | 8.46 (2.2×) | 44.72 | 46.8 |
| | + Fast-dLLM | 9.61 (2.8×) | 39.07 | 39.6 | 9.50 (2.5×) | 43.09 | 49.1 |
| | d²Cache | **11.68** (3.4×) | **27.68** | **40.7** | **12.67** (3.3×) | **27.17** | **50.8** |

## D.3 LIMITATIONS AND FUTURE WORK

Although d²Cache delivers substantial inference speedups across multiple models and datasets while maintaining comparable performance, several limitations have also emerged. Below we summarize these limitations and further outline potential directions for future work.

**Larger-scale dLLMs.** In this work, we closely follow recent representative practices (Wu et al., 2025; Liu et al., 2025) to evaluate d²Cache on LLaDA-8B (Nie et al., 2025) and Dream-7B (Ye et al., 2025). We note that, at this moment, LLaDA-8B and Dream-7B are the only publicly available dense dLLMs. As future dLLMs continue to scale up in depth, width, and context length, their bidirectional attention patterns will become even more costly to maintain during decoding. This trend further highlights the importance of more effective caching schemes. We view extending d²Cache to larger-scale dLLMs—together with more effective caching schemes—as a promising direction for future work, especially as model sizes and application demands continue to explode.

**Lightweight variants of attention rollout.** Although attention rollout is not a performance bottleneck in our d²Cache, its cost can become significant when it is applied to larger-scale models. More efficient approximations are therefore desirable. We evaluate a lightweight variant that computes rollout using only the first five layers on Dream-Inst and four datasets. As shown in Table 4, reducing the rollout depth from 28 to 5 yields a slight improvement in decoding speed while noticeably degrading performance on math reasoning tasks (GSM8K and Math-500); in contrast, code-generation tasks (HumanEval and MBPP) exhibit minimal performance loss. Designing lightweight rollout variants that can identify key tokens still remains an important direction for future work.

**Alternative scoring functions for contextual contribution.** We currently employ a Gaussian function to characterize how a masked token influences its surrounding context. While this approach performs well empirically, more context-adaptive formulations may further enhance performance.

Table 6: Comprehensive evaluation results on LLaDA-Inst (Nie et al., 2025) and Dream-Inst (Ye et al., 2025) with semi-AR parallel decoding. **Bold** numbers indicate the best results and green texts denote the speedup ratios.

| Dataset | Method | LLaDA-Inst | | | Dream-Inst | | |
|---|---|---|---|---|---|---|---|
| | | Throughput ↑ | Latency(s) ↓ | Score ↑ | Throughput ↑ | Latency(s) ↓ | Score ↑ |
| **GSM8K** 
 *4-shot* 
 Gen. Len. = 256 | Vanilla | 2.77 (1.0×) | 110.26 | **77.6** | 2.62 (1.0×) | 85.94 | **76.7** |
| | Parallel | 8.53 (3.1×) | 33.95 | **77.6** | 13.93 (5.3×) | 21.68 | 74.2 |
| | + dLLM-Cache | 25.62 (9.2×) | 10.26 | 77.0 | 36.00 (13.7×) | 8.21 | 74.3 |
| | + Fast-dLLM | 25.15 (9.1×) | 10.65 | **77.6** | 32.75 (12.5×) | 8.35 | 74.1 |
| | + d²Cache | **38.16** (13.8×) | **7.26** | 76.9 | **46.69** (17.8×) | **6.13** | 75.7 |
| **MBPP** 
 *3-shot* 
 Gen. Len. = 512 | Vanilla | 2.48 (1.0×) | 199.90 | 14.4 | 2.73 (1.0×) | 182.78 | 52.0 |
| | Parallel | 29.92 (12.1×) | 19.61 | 38.8 | 38.06 (13.9×) | 14.95 | 51.6 |
| | + dLLM-Cache | 72.52 (29.2×) | 7.94 | 38.0 | 89.31 (32.7×) | 6.11 | 51.8 |
| | + Fast-dLLM | 43.29 (17.5×) | 12.44 | 38.4 | 51.04 (18.7×) | 10.40 | 52.4 |
| | + d²Cache | **119.81** (48.3×) | **4.64** | **40.2** | **108.39** (39.7×) | **5.11** | **52.8** |
| **HumanEval** 
 *0-shot* 
 Gen. Len. = 512 | Vanilla | 4.99 (1.0×) | 105.76 | 45.1 | 4.39 (1.0×) | 114.86 | 56.7 |
| | Parallel | 15.74 (3.2×) | 37.63 | 45.1 | 39.78 (9.1×) | 20.53 | 51.8 |
| | + dLLM-Cache | 27.88 (5.6×) | 20.54 | **48.2** | 48.50 (11.0×) | 14.75 | 53.7 |
| | + Fast-dLLM | 25.14 (5.0×) | 21.76 | 43.3 | 48.94 (11.1×) | 13.67 | **57.3** |
| | + d²Cache | **48.39** (9.7×) | **11.89** | 46.6 | **104.40** (23.8×) | **6.30** | **57.3** |
| **Math-500** 
 *4-shot* 
 Gen. Len. = 256 | Vanilla | 3.08 (1.0×) | 82.51 | 38.4 | 3.51 (1.0×) | 71.05 | **45.2** |
| | Parallel | 8.80 (2.9×) | 31.99 | 38.4 | 12.75 (3.6×) | 23.99 | 44.4 |
| | + dLLM-Cache | 17.90 (5.8×) | 15.53 | 37.8 | 25.16 (7.2×) | 12.06 | 43.0 |
| | + Fast-dLLM | 24.49 (8.0×) | 11.01 | 37.4 | 28.68 (8.2×) | 9.73 | 43.4 |
| | + d²Cache | **34.69** (11.3×) | **8.03** | **38.6** | **37.83** (10.8×) | **7.74** | 42.6 |
| **GPQA** 
 *0-shot* 
 Gen. Len. = 256 | Vanilla | 6.14 (1.0×) | 43.34 | 25.2 | 6.43 (1.0×) | 41.14 | 30.1 |
| | Parallel | 44.61 (7.3×) | 15.98 | 25.7 | 128.85 (20.0×) | 2.87 | 31.3 |
| | + dLLM-Cache | 50.17 (8.2×) | 10.78 | 28.4 | **174.83** (27.2×) | **2.06** | 33.3 |
| | + Fast-dLLM | 42.66 (6.9×) | 10.13 | 25.9 | 136.28 (21.2×) | 2.31 | **34.6** |
| | + d²Cache | **89.03** (14.5×) | **6.89** | **28.8** | 162.95 (25.3×) | 2.08 | 32.8 |
| **MMLU-Pro** 
 *5-shot* 
 Gen. Len. = 256 | Vanilla | 1.76 (1.0×) | 152.62 | **37.5** | 2.15 (1.0×) | 126.31 | 47.9 |
| | Parallel | 8.82 (5.0×) | 58.86 | 37.2 | 17.87 (8.3×) | 27.12 | 47.8 |
| | + dLLM-Cache | 19.76 (11.2×) | 18.95 | 35.2 | 41.83 (19.5×) | 8.88 | **48.9** |
| | + Fast-dLLM | 19.46 (11.1×) | 16.07 | 37.1 | 35.46 (16.5×) | 8.94 | 47.1 |
| | + d²Cache | **28.85** (16.4×) | **13.24** | 35.1 | **54.56** (25.4×) | **6.42** | 46.1 |
| **AVG** | Vanilla | 3.54 (1.0×) | 115.73 | 39.7 | 3.64 (1.0×) | 103.68 | 51.4 |
| | Parallel | 19.40 (5.5×) | 33.00 | 43.8 | 41.87 (11.5×) | 18.52 | 50.2 |
| | + dLLM-Cache | 35.64 (10.1×) | 14.00 | 44.1 | 69.27 (19.0×) | 8.68 | 50.8 |
| | + Fast-dLLM | 30.03 (8.5×) | 13.68 | 43.3 | 55.53 (15.3×) | 8.90 | **51.5** |
| | + d²Cache | **59.82** (16.9×) | **8.66** | **44.4** | **85.80** (23.6×) | **5.63** | 51.2 |

# E ADDITIONAL EXPERIMENTAL RESULTS

## E.1 EXPERIMENTAL RESULTS ON THE BASE VARIANTS

In addition to the Instruct variants of LLaDA-8B (Nie et al., 2025) and Dream-v0-7B (Ye et al., 2025), we also conduct experiments on their Base variants, which are denoted as LLaDA-Base and Dream-Base, respectively. As shown in Table 5, our d²Cache consistently outperforms other approximate KV cache methods in terms of both average inference efficiency and accuracy across six datasets.

## E.2 EXPERIMENTAL RESULTS UNDER PARALLEL DECODING SETTINGS

To enable a fair comparison across all methods and to verify the generalizability of d²Cache under alternative decoding strategies, we evaluate all approaches using the parallel decoding strategy, where the threshold is set to 0.9 following Wu et al. (2025). As shown in Table 6, our method achieves up to 48.3× acceleration over the single-token-per-step baseline while maintaining performance comparable to all other baselines, which clearly demonstrates the broad applicability of d²Cache.

Table 7: Performance comparison on Dream-Inst with a generation length of 1024.

| Dataset | Method | Throughput (tokens/s) ↑ | Latency(s) ↓ | Score ↑ | Memory (GB) ↓ |
|---|---|---|---|---|---|
| **GSM8K** 
 *4-shot* 
 Gen. Len. = 1024 | Vanilla | 1.54 (1.0×) | 671.35 | 68.46 | 19.26 |
| | Fast dLLM | 4.18 (2.7×) | 245.29 | 67.85 | 19.39 |
| | dLLM-Cache | 3.33 (2.2×) | 308.62 | **68.76** | 20.28 |
| | d²Cache | **8.58** (5.6×) | **119.69** | 66.29 | 19.35 |
| **Math-500** 
 *4-shot* 
 Gen. Len. = 1024 | Vanilla | 1.89 (1.0×) | 541.04 | **43.6** | 19.15 |
| | Fast dLLM | 4.43 (2.3×) | 231.45 | 42.6 | 19.27 |
| | dLLM-Cache | 2.75 (1.5×) | 373.1 | 40.4 | 20.2 |
| | d²Cache | **9.55** (5.1×) | **107.29** | 41.2 | 19.28 |
| **HumanEval** 
 *0-shot* 
 Gen. Len. = 1024 | Vanilla | 2.62 (1.0×) | 393.47 | 56.71 | 19.06 |
| | Fast dLLM | 4.77 (1.8×) | 214.5 | 58.53 | 19.16 |
| | dLLM-Cache | 3.03 (1.2×) | 338.21 | 60.97 | 19.79 |
| | d²Cache | **11.74** (4.5×) | **87.64** | **64.02** | 19.14 |
| **MBPP** 
 *3-shot* 
 Gen. Len. = 1024 | Vanilla | 1.95 (1.0×) | 526.86 | 52.8 | 19.12 |
| | Fast dLLM | 4.45 (2.3×) | 229.91 | 52.4 | 19.23 |
| | dLLM-Cache | 4.59 (2.4×) | 223.05 | 54.2 | 19.94 |
| | d²Cache | **9.76** (5.0×) | **105.05** | **56.4** | 19.24 |
| **AVG** | Vanilla | 2.00 (1.0×) | 533.18 | 55.39 | 19.15 |
| | Fast dLLM | 4.46 (2.2×) | 230.29 | 55.35 | 19.26 |
| | dLLM-Cache | 3.43 (1.7×) | 310.75 | 56.08 | 20.05 |
| | d²Cache | **9.91** (5.0×) | **104.92** | **56.98** | 19.25 |

### E.3 EXPERIMENTAL RESULTS UNDER LONG-CONTEXT SETTINGS

To further assess our method's performance under long-context settings, we further evaluate our method on Dream-Inst under with a longer generation length 1024. As shown in Table 7, we observe that other methods—due to their coarse-grained nature—experience severely degraded acceleration as the context length increases. In contrast, d²Cache maintains substantial speedups without performance loss even in long-context scenarios, owing to its fine-grained two-stage token selection. These results demonstrate that d²Cache also performs well in long contexts.

### E.4 MORE VISUALIZATION RESULTS ON ATTENTION ROLLOUT

In this section, we present additional examples of attention rollout corresponding to the sample used in Figure 3. As shown in Figure 8, the attention pattern also aligns with our findings in Section 3.3.

### E.5 MORE VISUALIZATION RESULTS ON KV STATE DYNAMICS

To substantiate our findings in Section 3.2, we visualize additional KV state dynamics. In Figure 9, which visualizes the trajectories of the key and value states of the same masked token during decoding, both are closely aligned in both trajectory shape and magnitude, and both exhibit the same gradual–rapid–stable dynamic pattern. This result suggests that, for both key and value states, it is sufficient to update them only during the rapid-change phase, where these KV states can be safely cached for reuse during the other two phases. We hypothesize that this rapid change arises because tokens are particularly sensitive to changes in their local context. Specifically, at step $t$, if a masked token $x_t^i$ is located near another masked token $x_t^j$ that is decoded, then at step $t+1$ the embedding of $x_t^j$ changes from `[MASK]` to the embedding of a concrete token. This provides $x_t^i$ with additional contextual information; the smaller the distance $|i-j|$, the more tightly constrained the context becomes, thereby substantially altering the model's representation of $x_t^i$. These observations motivate the introduction of distance-aware decay into the certainty density, as defined in Equation (3).

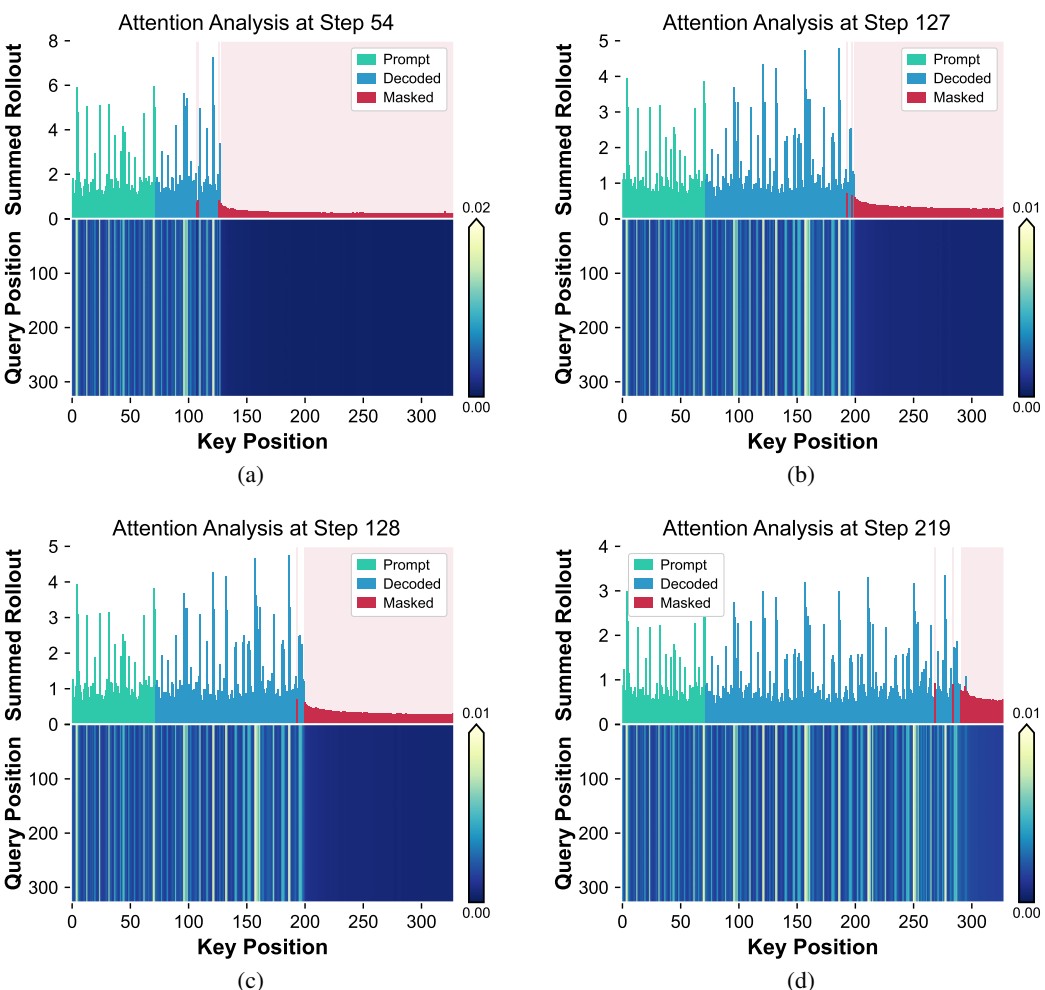

Figure 8: Visualization of attention rollout on LLaDA-Inst (Nie et al., 2025) with GSM8K, which is generated using the same sample and configuration as in Figure 3.

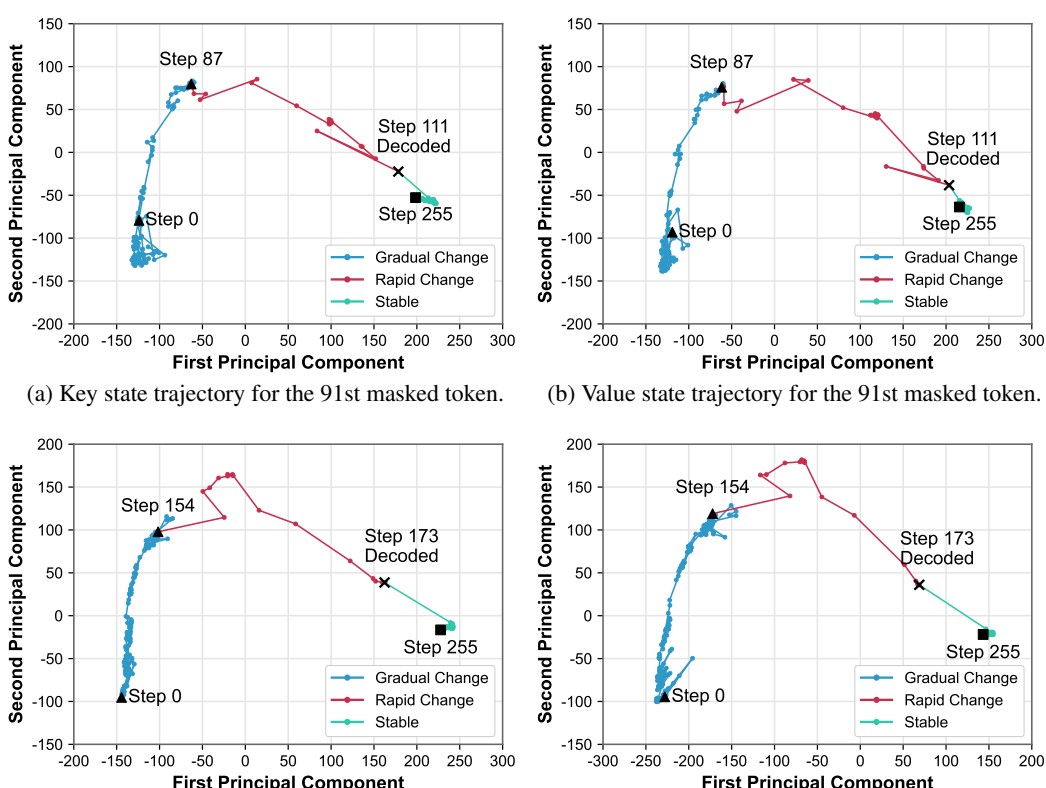

(a) Key state trajectory for the 91st masked token.

(b) Value state trajectory for the 91st masked token.

(c) Key state trajectory for the 186th masked token.

(d) Value state trajectory for the 186th masked token.

Figure 9: Visualization of PCA-projected trajectories of LLaDA-Inst on GSM8K, which are generated using the same sample and configuration as in Figure 2 (a).

