# OpenReview forum: "d$^2$Cache: Accelerating Diffusion-Based LLMs via Dual Adaptive Caching"
_ICLR.cc/2026/Conference — ICLR 2026 Poster_

### Official Review · Reviewer_E2JM · 2025-10-17

**Soundness:** 2
**Presentation:** 2
**Contribution:** 2
**Rating:** 2
**Confidence:** 4

**Summary:**

This paper proposes D2Cache, a training-free approximate key-value (KV) cache framework designed to accelerate inference in diffusion-based large language models (dLLMs). Unlike autoregressive models (ARMs), dLLMs rely on bidirectional attention and cannot directly reuse KV states efficiently.

D2Cache introduces a two-stage fine-grained token selection process:
1. Certainty Prior-Guided Selection: Selects masked tokens to update based on both their prediction confidence and local token density, yielding a more reliable quasi-left-to-right decoding order.
2. Attention-Aware Selection: Updates KV states only for tokens that receive higher attention weights, identified using attention rollout.

Through this adaptive caching mechanism, D2Cache reduces redundant computations and achieves significant speedups (3–4×) over baselines such as dLLM-Cache and Fast-dLLM, while maintaining or slightly improving task performance across math datasets.

**Strengths:**

1. The paper proposes a dual adaptive caching scheme that fine-tunes token-level updates for dLLMs. This is a clear step forward compared to previous coarse-grained cache methods like Fast-dLLM and dLLM-Cache.
2. D2Cache demonstrates up to 4× acceleration without significant performance loss. The method achieves competitive or better results than baselines across multiple reasoning and programming datasets.

**Weaknesses:**

1. The baseline (confidence-based decoding) performs much worse than the proposed certainty prior-guided method, even when acceleration isn’t applied. This large performance gap raises doubts about whether the comparison is fair or due to confounding factors. It would be useful to include random sampling or alternative decoding methods (e.g., semi-autoregressive decoding) for stronger baselines.

2. There’s ambiguity between speedup runs (Table 1) and default non-autoregressive runs (Table 2). Where Table 1 evaluates speedup under different methods, and Table 2 evaluates decoding schemes (confidence-based vs certainty-prior). However, both report exacly same performance scores, which makes it unclear whether the performance metrics are decoupled from the acceleration setting or derived from identical runs.
3. The experiments focus only on math and code-related tasks. It remains unclear how the proposed method generalizes to general tasks. For example, perplexity should be evaluated as a basic metric.

**Questions:**

1. Since the proposed decoding strategy (certainty prior-guided) enforces a quasi-left-to-right order, how does it differ from simply using an autoregressive model? If the pipeline behaves like a “quasi-AR” system, does the original advantage of dLLMs (e.g., bidirectional modeling, parallel decoding) still hold?
2. Were the baseline methods (dLLM-Cache, Fast-dLLM) tuned optimally for the same tasks? Given the performance gap, could hyperparameter choices or block sizes explain some of the observed differences?

---

> ### Author Response · Authors · 2025-11-20
> **Response to Reviewer E2JM (Part 1/3)**
>
> # Dear Reviewer E2JM,
> We sincerely appreciate your constructive feedback and valuable suggestions. We include extensive experiments and thorough discussions to address each of your concerns. Our detailed responses are as follows.
>
> ---
>
> ## R4W1: Potential Unfair Comparisons
> Our comparisons are fair. As you observed, the baseline (confidence-based decoding) performs poorly, which is consistent with the findings in the LLaDA paper [1]. In Appendix B.3, they report:
> > For instruct models, since our training includes a large number of |EOS| tokens, directly applying the lowest-confidence remasking strategy would result in an excessively high and unnatural occurrence of |EOS| tokens in the generated sentences.
>
> To alleviate this issue, semi-AR enforces a quasi left-to-right decoding order, which partially mitigates the EOS inflation problem, though [1] does not explain why semi-AR works. Through our analysis, we find that the key reason semi-AR is effective is that it implicitly prioritizes masked tokens closer to known tokens in decoding distance. Motivated by this insight, we propose a more direct solution—certainty prior-guided decoding.
>  We additionally compare against random sampling and include the Semi-AR (vanilla) results reported in Table 1 of [1]; the performance comparisons are shown in the table.
>
> ### **Decoding Strategy Comparison on Dream-Inst**
>
> | **Benchmarks** | **NAR** | **Random** | **Semi AR** | **NAR w/ Only CP** | **NAR w/ $d^2$Cache** |
> | :--- | :--- | :--- | :--- | :--- | :--- |
> | GSM8k | 51.6 | 32.9 | 76.7 | 78.1 | **78.2** |
> | Math | 3.2 | 13.4 | **45.2** | 43.6 | 44.6 |
> | HumanEval | 26.8 | 20.1 | 56.7 | 54.3 | **61.6** |
> | MBPP | 34.2 | 30.6 | 52 | 59.2 | **58** |
>
> Compared to semi-AR, certainty prior-guided decoding achieves similar performance, which is expected since both methods share the same objective: enforcing a decoding order that prefers masked tokens with richer contextual information. However, our method provides a clearer conceptual perspective on what constitutes a principled decoding order.
>
> ---
>
> ## R4W2: Experimental Settings in Table 1 and Table 2
>
> We apologize for the confusion. As stated in Line 722 of the main paper: *"all methods except d2Cache adopt block-wise semi-autoregressive (semi-AR) decoding with a block size of 32”*, whereas d2Cache uses the certainty prior-guided decoding proposed in Section 4.1. Therefore, Table 1 and Line 384 present identical performance scores.
>
> To further ensure fairness, we evaluate all methods on Dream-Inst under exactly the same decoding strategy (semi-AR decoding). Moreover, we set the decoding confidence threshold to 0.9—retaining all tokens with confidence above 0.9—which corresponds to the parallel decoding strategy in Fast dLLM [2]. The results show that our method achieves at most $39.7×$ acceleration over the single-token-per-step baseline, while maintaining comparable performance with all other baselines.
>
> ### **Performance on Dream-Inst with Semi-AR Decoding**
>
> | **Benchmarks** | **Metric** | **Vanilla** | **Parallel** | **Fast dLLM** | **dLLM-Cache** | **Ours** |
> | :--- | :--- | :--- | :--- | :--- | :--- | :--- |
> | **GSM8k** | Score | **76.65** | 74.22 | 74.07 | 74.29 | 75.74 |
> | | Speed | 2.62 | 13.93 (5.3x) | 32.75 (12.5x) | 36 (13.7x) | **46.69 (17.8x)** |
> | | Latency | 85.94 | 21.68 | 8.35 | 8.21 | **6.13** |
> | **Math** | Score | **45.2** | 44.4 | 43.4 | 43 | 42.6 |
> | | Speed | 3.51 | 12.75 (3.6x) | 28.68 (8.2x) | 25.16 (7.2x) | **37.83 (10.8x)** |
> | | Latency | 71.05 | 23.99 | 9.73 | 12.06 | **7.74** |
> | **HumanEval** | Score | 56.71 | 51.82 | 57.31 | 53.65 | **57.32** |
> | | Speed | 4.39 | 39.78 (9.1x) | 48.94 (11.1x) | 48.5 (11.0x) | **104.4 (23.8x)** |
> | | Latency | 114.86 | 20.53 | 13.67 | 14.75 | **6.3** |
> | **MBPP** | Score | 52 | 51.6 | 52.4 | 51.8 | **52.8** |
> | | Speed | 2.73 | 38.06 (13.9x) | 51.04 (18.7x) | 89.31 (32.7x) | **108.39 (39.7x)** |
> | | Latency | 182.78 | 14.95 | 10.4 | 6.11 | **5.11** |

---

> ### Author Response · Authors · 2025-11-20
> **Response to Reviewer E2JM (Part 2/3)**
>
> ## R4W3: General Tasks and Perplexity
>
> **General tasks**. Following the reviewer's suggestion, we evaluate all methods on MMLU and GPQA. As shown in the table, our method consistently outperforms the baselines in speed while achieving performance comparable to vanilla decoding.
>
> ### **Performance of General Tasks on Dream-Inst**
> | Benchmarks | Metric | Vanilla | Fast-dLLM | dLLM-Cache | Ours |
> | :--- | :--- | :--- | :--- | :--- | :--- |
> | GPQA  | Perf | 30.13 | **34.59** | 31.02 | 31.47 |
> | | Speed | 6.43 | 11.75 (1.8x) | 10.91 (1.7x) | **14.65 (2.3x)**  |
> | MMLU  | Perf | **47.92** | 45.92 | 46.45 | 46.79 |
> | | Speed | 2.15 | 9.04 (4.2x) | 7.12 (3.3x) | **10.12 (4.7x)**  |
>
> **Perplexity.** Although perplexity is widely used in conventional autoregressive models, it is not a suitable metric for dLLMs. Let us explain this step by step. In an autoregressive model, when predicting the $t$-th token $w_t$, the model conditions on the past tokens $W_{<t}:=(w_1,\dots,w_{t-1})$. The perplexity PPL of a sentence is defined as $\mathrm{PPL}=\exp\big(-\tfrac{1}{N}\mathrm{LL}\big)$, where the sentence log-likelihood LL is
> $\mathrm{LL}:=\log P_{\mathrm{LM}}(W)=\sum_{t=1}^{|W|}\log P_{\mathrm{LM}}(w_t\mid W_{<t})$
>
> However, dLLMs do not perform generation in a strictly autoregressive fashion, so the conditional terms $P(w_t\mid W_{<t})$ are not directly available. To address this in masked-language-model (MLM) settings, Julian et al. [3] proposed the pseudo-likelihood (PLL) to estimate sentence likelihoods for MLMs. The PLL is defined as
> $\mathrm{PLL} := \sum_{t=1}^{|W|}\log P_{\mathrm{MLM}}(w_t\mid W_{\setminus t})$
>  i.e., for each token we mask that token and use the model to predict its probability given the remaining context. This metric has been widely adopted in works such as LLaDA and Dream. Crucially, computing PLL requires evaluation on ground-truth data. For example, in multiple-choice tasks one concatenates the question and the candidate option, masks every token in the option one at a time, and computes the PLL. This procedure does not involve generation, and therefore it does not make use of KV caching.
>
> ---
>
> ## R4Q1: Difference Between Quasi-AR and AR Systems
> We would like to clarify that our method does not enforce quasi-left-to-right generation for dLLMs. Instead, dLLMs themselves tend to perform better in a quasi-left-to-right mode, especially on sequential reasoning tasks [4]. In practice, semi-AR decoding aligns with this mode by constraining the model to generate approximately from left to right, which can largely preserve reasoning performance, as shown in [4].
>
> Compared to strictly autoregressive models, however, dLLMs still retain greater flexibility during generation. For example, when the model encounters a position where all next-token candidates have low confidence, an AR model has to commit to a choice. In contrast, a dLLM can decode further positions and make the decision only after the extended context provides sufficient evidence, thereby leveraging its non-AR modeling capacity. A concrete example is pronoun resolution in ambiguous contexts. Suppose the prompt is:
> *“Alice thanked Mary because ___ had helped with the project.”*
>
>  At the blank position, an AR model must immediately choose between “she” and “Alice”, even though the correct antecedent is unclear without further context. A dLLM, however, can tentatively fill in both possibilities, continue decoding additional positions, and use the extended context to decide whether the sentence is going to continue as
> *“… she had provided key data,”*
>  or
> *“… Alice needed assistance,”*
>  before committing to the final token.
>
>  Additionally, even when using quasi-AR decoding, each masked token still attends to the entire context (unlike in ARM, where tokens attend only to previous positions), making the original advantages of dLLMs (e.g., bidirectional modeling, parallel decoding) still hold.
>
> ---
>
> ## R4Q2: Optimality of Hyperparameters in Baseline Methods
>
> The hyperparameters used for all baselines follow the best settings reported in their respective papers, as documented in Appendix B. For Fast dLLM, we follow Figure 4 in [2]; for dLLM-cache, we follow Tables 1 & 2 in [5]. It is worth noting that these methods can only be tuned to achieve lossless performance relative to vanilla decoding, whereas d2Cache surpasses them in both average speed and average performance.

---

> ### Author Response · Authors · 2025-11-20
> **Response to Reviewer E2JM (Part 3/3)**
>
> ## Reference
>
> [1] Nie S, Zhu F, You Z, et al. Large language diffusion models[J]. arXiv preprint arXiv:2502.09992v2, 2025.
>
> [2] Wu C, Zhang H, Xue S, et al. Fast-dllm: Training-free acceleration of diffusion llm by enabling kv cache and parallel decoding[J]. arXiv preprint arXiv:2505.22618, 2025.
>
> [3] Salazar J, Liang D, Nguyen T Q, et al. Masked language model scoring[C]//Proceedings of the 58th annual meeting of the association for computational linguistics. 2020: 2699-2712.
>
> [4] Huang P, Liu S, Liu Z, et al. Pc-sampler: Position-aware calibration of decoding bias in masked diffusion models[J]. arXiv preprint arXiv:2508.13021, 2025.
>
> [5] Liu Z, Yang Y, Zhang Y, et al. dllm-cache: Accelerating diffusion large language models with adaptive caching[J]. arXiv preprint arXiv:2506.06295, 2025.

---

> ### Comment · Reviewer_E2JM · 2025-11-21
>
> Thanks for your comment. In this part, my further opinions are as follows:
> 1. Since SemiAR achieves a satisfactory score and is applied in the expertments in Table 1, the comparsion in Table 2 is quite misleading. At least the following SemiAR results should be added into Table 2. Even though certainty prior-guided decoding proposes a conceptual perspective, it also shares the spirit of SemiAR. Since your contribution is more on acceleration rather than improving performance, it is not appropriate to directly compare your method with weaker baselines.
>
> 2. As a discussion, do you think it is better to compare AR inference speed as a candidate for measuring speed-up? Vanilla dLLM's inference is well-known not efficient. Since dLLM is still not widely accepted as industry-level LLM framework, I personally believe that only achiving acceleration on dLLM is not enough.

---

> ### Author Response · Authors · 2025-11-21
> **Response to New Comment by Reviewer E2JM (Part 1/2)**
>
> ## R4Opinion 1: Misleading Comparison among Decoding Strategies
>
> We sincerely appreciate the reviewer's constructive suggestion regarding the comparison in Table 2. We respectfully agree that the concern is indeed reasonable: since semi-AR already achieves strong performance and is included in Table 1, ignoring its results in Table 2 may give the impression that the comparison is unfair and incomplete.
>
> We would like to clarify that the purpose of Table 2 is not to intentionally compare against weak baselines to exaggerate the effectiveness of our decoding strategy. As the reviewer correctly points out, our primary contribution lies in accelerating dLLM inference rather than surpassing strong baselines in accuracy. Our intention in Table 2 is to demonstrate that the proposed decoding strategy provides a more reasonable and effective alternative to the default NAR setting as used in current dLLMs. Although semi-AR and certainty prior–guided decoding share a similar intuition, we observe that only the combination of certainty prior–guided decoding and d²Cache delivers the best performance among all evaluated configurations, as summarized in the following table.
>
> ### **Dream-Inst Performance**
>
> | Benchmarks | Vanilla (semi-AR) | Parallel (semi-AR) | NAR w/ CP | semi-AR w/ $d^2$Cache | NAR w/ $d^2$Cache |
> | :--- | :--- | :--- | :--- | :--- | :--- |
> | **GSM8k** | 76.65 | 74.22 | 78.1 | 75.74 | 78.24 $(+2.5)$ |
> | **Math** | 45.2 | 44.4 | 43.6 | 42.6 | 44.6 $(+2)$ |
> | **HumanEval** | 56.71 | 51.82 | 54.3 | 57.32 | 61.58 $(+4.26)$ |
> | **MBPP** | 52 | 51.6 | 59.2 | 52.8 | 58 $(+5.2)$ |
>
> We would like to thank the reviewer again for the helpful suggestion, which improves the clarity and completeness of our empirical evaluation.
>
> ---
>
> ## R4Opinion 2: Discussion on Inference Speed of AR Models and dLLMs.
>
> We sincerely appreciate the reviewer’s thoughtful perspective regarding the practical value of dLLMs. It is true that, in current industrial scenarios, dLLMs still lag behind AR models, particularly in terms of inference efficiency. This mainly results from the fact that the diffusion-based paradigm for language generation only recently begins to scale up [1,2] and starts to demonstrate performance comparable to AR models. In this sense, directly comparing the inference efficiency of dLLMs with that of AR models can be somewhat unfair at the current stage. In practice, AR models can benefit from many years of infrastructure development and a large body of mature acceleration techniques, such as KV cache mechanisms [3]  and speculative decoding works [4], which have been extensively refined and adopted in real-world industrial scenarios. In contrast, dLLMs are only beginning to build such an optimization infrastructure.
>
> However, we would like to offer a broader view based on the historical trajectory of diffusion models in other domains. The development of diffusion models for images provides a highly relevant reference. As seen in the literature, early image diffusion models (e.g.,  DDPM [5]) deliver impressive generative quality but suffer from extremely slow sampling. Nevertheless, these early works are far from “insufficient”; instead, they lay essential conceptual foundations that directly enable the rapid emergence of vastly more efficient successors—including consistency models [6]—that ultimately achieve over three orders of magnitude improvement within only a few years. The research community widely recognizes that these initial methods play an important role in catalyzing subsequent breakthroughs.
>
> In the same spirit, the current generation of dLLMs remains at a comparable early stage. Their performance and efficiency continue to evolve, and the surrounding infrastructure advances at a rapid pace. Therefore, accelerating dLLM inference—even if the current efficiency does not yet match AR models—serves as an essential step in this iterative process. Such research aims not merely to match AR-level inference efficiency today, but to shape the technological trajectory that may fundamentally influence how future LLMs are designed and deployed. We further believe that, as the training recipes, model architectures, and inference optimizations for dLLMs evolve, the performance and efficiency of dLLMs will continue to improve, gradually reach, and eventually surpass AR models.

---

> > ### Author Response · Authors · 2025-11-21
> > **Response to New Comment by Reviewer E2JM (Part 2/2)**
> >
> > ## Reference
> >
> > [1] Nie S, Zhu F, You Z, et al. Large language diffusion models[J]. arXiv preprint arXiv:2502.09992v2, 2025.
> >
> > [2] Ye J, Xie Z, Zheng L, et al. Dream 7b: Diffusion large language models[J]. arXiv preprint arXiv:2508.15487, 2025.
> >
> > [3] Zhang Z, Sheng Y, Zhou T, et al. H2o: Heavy-hitter oracle for efficient generative inference of large language models[J]. Advances in Neural Information Processing Systems, 2023, 36: 34661-34710.
> >
> > [4] Leviathan Y, Kalman M, Matias Y. Fast inference from transformers via speculative decoding[C]//International Conference on Machine Learning. PMLR, 2023: 19274-19286.
> >
> > [5] Ho J, Jain A, Abbeel P. Denoising diffusion probabilistic models[J]. Advances in neural information processing systems, 2020, 33: 6840-6851.
> >
> > [6] Lu C, Song Y. Simplifying, stabilizing and scaling continuous-time consistency models[J]. arXiv preprint arXiv:2410.11081, 2024.

---

> ### Comment · Reviewer_E2JM · 2025-11-23
>
> Dear authors:
>
> Thanks for your following discussion. My doubts about the paper's motivation have been solved. Therefore, I will increase my score to 4.
>
> The reason that I still lean to reject the paper is not on the techniqual details, but on the broad value and influence to the whole community:
> 1. The acceleration method introduced in this paper looks still an extension based on previous fast inference series for dLLMs to me. Even though the acceleration ratio looks promising compared with vanilla inference, the advantage compared with other baselines is not signifiant enough.
> 2. The metrics introduced in 4.1 and 4.2 are more like hand-crafted equations. Even though the experiment works well, the extension for future work is limited.
>
> Thanks for the discussion again. Since there is no misunderstanding about the paper itself, I will keep my opinion if no fundamental change happens.

---

> ### Author Response · Authors · 2025-11-23
> **Response to New Comment by Reviewer E2JM (Part 1/1)**
>
> # Dear Reviewer E2JM,
>
> We would like to thank you for your follow-up comments. We sincerely appreciate your engagement with our work, and we would like to further clarify the two reasons you mentioned.
>
> ---
>
> ## R4Reason 1: The advantage compared with other baselines is not significant enough.
>
> We respectfully disagree with this point. Under exactly the same decoding strategy, our method achieves **1.8x and 1.5x inference speedups** over two representative baselines (i.e., Fast-dLLM and dLLM-Cache), while maintaining **comparable or even better performance** (see R4W2). We acknowledge that the notion of “significance” is often subjective; however, according to the [ICLR 2026 reviewer guide](https://iclr.cc/Conferences/2026/ReviewerGuide):
> > Q: If a submission does not achieve state-of-the-art results, is that grounds for rejection?
> >
> > A: No, a lack of state-of-the-art results does not by itself constitute grounds for rejection. Submissions bring value to the ICLR community when they convincingly demonstrate new, relevant, impactful knowledge. Submissions can achieve this without achieving state-of-the-art results.
>
>
> This reflects the core spirit of the ICLR community: what matters most is whether the submission demonstrates new, relevant, and impactful insights, rather than achieving state-of-the-art performance (In our case, our work does achieve competitive or state-of-the-art performance).
>
> In previous relevant methods (e.g., Fast-dLLM and dLLM-Cache), the motivation has been largely the same. That is, adjacent decoding steps in dLLMs produce similar representations, so the KV cache can be refreshed at **fixed** intervals to reduce computation. In contrast, our work explores token-level dynamics and attention patterns, enabling the model itself to **adaptively** decide which tokens actually need to be cached or refreshed, based on both input-dependent and model-internal representations. Moreover, we further characterize the decoding order of dLLMs and introduce a more conceptual perspective (i.e., certainty prior–guided decoding), which goes beyond the heuristic nature of semi-autoregressive methods. Hence, our work indeed provides new perspectives and conceptual understanding, rather than merely serving as an extension of previous relevant methods.
>
> ---
>
> ## R4Reason 2: The extension for future work is limited.
>
> We believe that our work naturally opens several promising directions for future research:
>
> - **Section 4.1**: We use a Gaussian function to model the influence of a masked token on its neighboring context. While this formulation works well in our experiments, as you have noted, more context-dependent formulations may yield better results.
> - **Section 4.2**: Our current approach identifies pivotal tokens based on their attention rollout values. As suggested by Reviewers 4Cqp and 1oge, incorporating token sparsification techniques to automatically discover pivotal tokens could be highly beneficial, especially as dLLMs continue to scale up.
> - **Adaptive selection of (k, p)**: In R1Q2, we have explored lightweight mechanisms for adaptively determining the values of ($k$, $p$). Such adaptive strategies will help address the issue highlighted in R3Q3. That is, when a dLLM performs poorly on a task, its attention or confidence may not be reliable. In these cases, it becomes necessary to increase $p$ or $k$ to compensate for this limitation. Exploring such directions can make our method more flexible and efficient.
>
> ---
>
> Finally, we sincerely appreciate your acknowledgment of our work and your thoughtful engagement. Thank you!

---

### Official Review · Reviewer_1oge · 2025-10-20

**Soundness:** 3
**Presentation:** 2
**Contribution:** 2
**Rating:** 4
**Confidence:** 3

**Summary:**

To solve diffusion-based LLMs’ (dLLMs) inference inefficiency (cannot use standard KV caches), the paper proposes d2Cache, a training-free KV cache framework. It leverages two insights: (1) masked tokens’ KV states only need updates in the "rapid-change" phase; (2) dLLM attention concentrates on a small token subset. d2Cache uses two-stage selection (certainty prior for masked tokens, attention rollout for prompt/decoded tokens) and guided decoding to avoid premature termination. Experiments on 2 dLLMs (LLaDA-8B/Dream-7B) and 4 benchmarks show 3.2×–4.0× speedup over vanilla dLLMs, outperforming baselines while improving accuracy.

**Strengths:**

1. Empirically grounded design with measurable evidence: The framework  relies on fine-grained analyses of dLLM behavior. For example, PCA visualizations confirm masked tokens’ three-phase KV evolution (e.g., LLaDA-8B-Inst/GSM8K shows rapid changes in steps 64–98), and attention rollout proves 80% of attention concentrates on 20% of prompt/decoded tokens. These insights show the "phase-based updates" and "high-attention selection" core strategies.
2. Most dLLM efficiency methods trade accuracy for speed, but d2Cache improves both. Its certainty prior-guided decoding mitigates vanilla dLLMs’ "U-shaped" bias (tokens at sequence boundaries decoded first, leading to premature, boosting accuracy while accelerating inference. For instance, on Dream-Inst/HumanEval, it raises throughput from 4.39 to 14.06 and accuracy from 56.7 to 61.6.

**Weaknesses:**

1. The paper only tests 7B/8B parameter models and sequences ≤512 tokens. For larger models (e.g., 13B/70B dLLMs), the computational overhead of attention rollout may scale sharply with model size, offsetting speed gains. It is interesting to discuss the proposed method on the experiments at scale.
2. The paper ignores scenarios where d2Cache underperforms. For example, on LLaDA-Inst/MBPP, its accuracy (12.4) is lower than Fast-dLLM (13.8) and dLLM-Cache (12.8)， although the proposed method is better at throughput and latency. It is interesting to explain whether code generation requires more frequent masked token updates (k=32 is too small), attention rollout misses code-specific critical tokens (e.g., function call syntax), or the certainty prior fails to capture code structure dependencies.

**Questions:**

1. Scalability to large models: How does d2Cache’s performance (speedup, accuracy) change when applied to 13B/70B dLLMs?  Can lightweight optimizations (e.g., sparse attention rollout, only using top-5 layers) reduce this overhead without losing accuracy?
2. Adaptation to long sequences: For sequences longer than 512 tokens (e.g., 1k PubMed abstracts, 2k legal documents), how should the Gaussian function’s σ be adjusted? Does a dynamic σ (e.g., σ = sequence_length × 0.02) maintain accuracy while avoiding excessive updates?
3. Failure case root cause: On LLaDA-Inst/MBPP, why does d2Cache’s accuracy (12.4) lag behind Fast-dLLM (13.8)? Are there specific code patterns (e.g., nested loops, import statements, function definitions) where the two-stage selection fails?

---

> ### Author Response · Authors · 2025-11-20
> **Response to Reviewer 1oge (Part 1/2)**
>
> # Dear Reviewer 1oge,
> We sincerely appreciate your constructive feedback and valuable suggestions. We have included extensive experiments and thorough discussions to address each of your concerns. Our detailed responses are as follows.
>
> ---
>
> ## R3W1 & R3Q1: Larger-Scale Models.
>
> Diffusion-based large language models (dLLMs) are still an emerging research direction, and only successfully scaling to 7 to 8 billion models this year. For example, LLaDA [1] was released in February this year and Dream [2] in May. A more complete timeline can be found in Figure 1 of the recent survey report [3]. Therefore, as you mentioned, the open-source community currently lacks larger-scale intensive models (e.g., 17B/70B and above). This is also why all existing dLLM caching work (e.g., Fast-dLLM and dLLM-Cache) focuses on LLaDA-8B and Dream-7B. We fully agree that evaluating larger-scale models would be very valuable and we would be happy to include them in our research once they become available.
>
> ---
>
> ## R3W1 & R3Q2: Longer Sequence.
>
> You mentioned that our paper only evaluates scenarios with sequences of length ≤512 tokens. In fact, the values 256 and 512 in the paper refer to generation lengths, not full sequence lengths. For example, in GSM8K (4-shot), the average prompt length is approximately 800 tokens; with an additional 256 generated tokens, the total sequence length is roughly 1k.
>
> To further assess our method’s performance under long-context settings, we further evaluate our method on Dream-Inst under (1) longer generation lengths (i.e., 1024 tokens) and (2) longer prompts (i.e., 1.4k  tokens).
> 1. Longer generation length (1024).
> We evaluate the performance of d2Cache on the four datasets used in our main paper.
> 2. Longer prompts (1.4k).
> We evaluate all methods on the MMLU-Pro (5-shot) dataset, where the average prompt length consists of 1.4k tokens.
>
> ### **Long Context Performance on Dream-Inst**
>
> | **Benchmarks** | **Metric** | **Vanilla** | **Fast dLLM** | **dLLM-Cache** | **Ours** |
> | :--- | :--- | :--- | :--- | :--- | :--- |
> | **GSM8k (Gen. Len = 1024)** | Score | 68.46 | 67.85 | **68.76** | 66.29 |
> | | Speed | 1.54 | 4.18 (2.7x) | 3.33 (2.2x)| **8.58 (5.6x)** |
> | | Latency | 671.35 | 245.29 | 308.62 | **119.69** |
> | **Math (Gen. Len = 1024)** | Score | **43.6** | 42.6 | 40.4 | 41.2 |
> | | Speed | 1.89 | 4.43 (2.3x) | 2.75 (1.5x)| **9.55 (5.1x)**  |
> | | Latency | 541.04 | 231.45 | 373.1 | **107.29** |
> | **HumanEval (Gen. Len = 1024)** | Score | 56.71 | 58.53 | 60.97 | **64.02** |
> | | Speed | 2.62 | 4.77 (1.8x) | 3.03 (1.2x)| **11.74 (4.5x)**  |
> | | Latency | 393.47 | 214.5 | 338.21 | **87.64** |
> | **MBPP (Gen. Len = 1024)** | Score | 52.8 | 52.4 | 54.2 | **56.4** |
> | | Speed | 1.95 | 4.45 (2.3x)| 4.59 (2.4x)| **9.76 (5.0x)**  |
> | | Latency | 526.86 | 229.91 | 223.05 | **105.05** |
> | **MMLU (Gen. Len = 256)** | Score | **47.92** | 45.92 | 46.45 | 46.79 |
> | | Speed | 2.15 | 9.04 (4.2x) | 7.12 (3.3x) | **10.12 (4.7x)**  |
> | | Latency | 126.31 | 27.69 | 34.09 | **25.77** |
>
> Across both settings, we observe that other methods—due to their coarse-grained nature—experience severely degraded acceleration as the context length increases. In contrast, d2Cache maintains substantial speedups without performance loss even in long-context scenarios, owing to its fine-grained two-stage token selection. These results clearly demonstrate the generality of d2Cache.
>
> ---
>
> ## R3Q2: Adjustment of $\sigma$ in Gaussian Function
>
> The $\sigma$ in Eq. (3) is used to control the decoding order: a smaller $\sigma$ makes the model behave more autoregressively, whereas a larger σ pushes it toward more non-autoregressive decoding. Therefore, $\sigma$ is not related to the frequency of cache updates.
> To clarify its effect, we evaluate the relationship between $\sigma$ and model performance on Dream-Inst under different generation lengths (256, 512, and 1024 tokens). The results are summarized in the table below. Across all settings, we observe that $\sigma=10$ typically yields the best performance, striking a good balance between AR-ness and decoding flexibility.
>
> ### **$\sigma$ Ablation on Dream-Inst, Humaneval**
> | GenLen \setminus \sigma$ | $\sigma=1$ | $\sigma=10$ | $\sigma=20$ | $\sigma=40$ | $\sigma=80$ |
> | :--- | :--- | :--- | :--- | :--- | :--- |
> | **Gen Len=256** | **62.2** | 61.6 | 58.54 | 57.93 | 54.88 |
> | **Gen Len=512** | 50 | **61.58** | 59.75 | 57.92 | 50 |
> | **Gen Len=1024** | 52.44 | **61.56** | 57.93 | 58.54 | 60.36 |

---

> ### Author Response · Authors · 2025-11-20
> **Response to Reviewer 1oge (Part 2/2)**
>
> ## R3W2 & R3Q3: Failure Case Root Cause on LLaDA-Inst/MBPP
>
> Compared with Fast-dLLM, which refreshes the cache according to a predefined update schedule, d2cache places more trust in the model’s own signals. It selects the update subset based on the model’s attention distribution or confidence. This naturally introduces a challenge. When a dLLM performs poorly on a task, its attention or confidence may not be reliable. In such cases, it becomes necessary to increase $p$ or $k$ to compensate for this limitation.
>
> Following the reviewer’s suggestion, we expand the range of $k$, but this did not lead to consistent performance gains. We therefore examine the effect of using a larger $p$. To do this, we analyze several MBPP examples where LLaDA-Inst gives correct answers without caching but fails when d2cache is enabled. By inspecting the tokens selected through attention rollout, we observed that trivial tokens, such as punctuation marks (e.g., "(", ")", ".") and newline characters ("\n"), often receive a disproportionately large amount of attention. With a small $p$, these tokens crowd out more important code-related tokens. After we normalize the input by replacing consecutive whitespace with a single space, the attention shift toward meaningful code-related tokens such as “assert”, “==”, “not”, “is”, and “True”. The model subsequently produced the correct output.
>
> ---
>
> ## R3W1 & R3Q1: Attention Rollout and its Lightweight Variants
>
> **Computational overhead of attention rollout.**
>  In our current experimental setup, attention rollout does not become a performance bottleneck. However, if future larger-scale models make attention rollout significantly more expensive during inference, more efficient strategies can be adopted—for instance, leveraging token sparsification techniques, as suggested by Reviewer 4Cqp. This is because Stage 2 in our method fundamentally requires selecting important tokens, and the underlying algorithm does not have to rely strictly on attention rollout.
>
> **Lightweight variants of attention rollout.**
> Thank you for the insightful suggestion. We evaluated the effect of computing attention rollout using only the first 5 layers across the four datasets in Dream-Inst, and the results are shown in the table below. Reducing the number of rollout layers from 32 to 5 yields a slight improvement in decoding speed, indicating that, under our current experimental setup, attention rollout is not a performance bottleneck.
>
> | **Benchmarks** | **Metric** | **First 5 layer rollout** | **Ours** |
> | :--- | :--- | :--- | :--- |
> | **GSM8k** | Score | 71.8 | **78.24** |
> | | Speed | 12.61 | 12.25 |
> | **Math** | Score | 40.2 | 44.6 |
> | | Speed | 13.99 | 13.8 |
> | **HumanEval** | Score | **62.2** | 61.58 |
> | | Speed | 14.2 | 14.06 |
> | **MBPP** | Score | 57.2 | **58** |
> | | Speed | 13.1 | 12.47 |
>
> In terms of model quality, however, we observe a substantial degradation on mathematical–reasoning tasks (GSM8K and MATH), especially on GSM8K, whereas code-generation tasks (HumanEval and MBPP) exhibit minimal performance change. These findings suggest that shallow rollout weakens the model’s ability to capture long-range or multi-step dependencies that are particularly important for mathematical reasoning.
>
> ---
>
> ## Reference
>
> [1] Nie S, Zhu F, You Z, et al. Large language diffusion models[J]. arXiv preprint arXiv:2502.09992, 2025.
>
> [2] Ye J, Xie Z, Zheng L, et al. Dream 7b: Diffusion large language models[J]. arXiv preprint arXiv:2508.15487, 2025.
>
> [3] Yu R, Li Q, Wang X. Discrete Diffusion in Large Language and Multimodal Models: A Survey[J]. arXiv preprint arXiv:2506.13759, 2025.

---

> ### Comment · Reviewer_1oge · 2025-11-24
> **Reply to authors**
>
> I appreciate your response to the questions about d2Cache’s large-model scalability, long-sequence adaptation, and failure case analysis, and I’ll keep my positive evaluation.

---

> ### Author Response · Authors · 2025-11-24
>
> We would like to thank the reviewer for your kind follow-up comment and for keeping your positive evaluation. From your latest comment, it appears that all your concerns have been fully addressed.
>
> Since your current score (i.e., 4) may still reflect a negative evaluation, we respectfully wonder whether it fully aligns with your stated positive evaluation. If you feel that our clarifications and additional experiments have resolved your concerns, we would be sincerely grateful if you could consider updating the score accordingly.
>
> Thank you again for your constructive feedback and for your support of our work.

---

### Official Review · Reviewer_KmGv · 2025-10-30

**Soundness:** 3
**Presentation:** 2
**Contribution:** 3
**Rating:** 6
**Confidence:** 3

**Summary:**

Authors proposes d2Cache, a training-free approximate cache that updates only a small subset of tokens’ KV states and reuses cached KV for the rest. Token selection is two-stage: Certainty-prior for masked tokens and attention-aware selection. The accuracy and efficieny are improved by this method.

**Strengths:**

1. KV cache for dLLM inference is new, where standard KV cache doesn’t apply.

2. Simple design with limited moving parts and clear hyperparameters.

3. consistent speedups and several accuracy improvements.

**Weaknesses:**

1. Please explore hyperparameters of k  and p. these likely trade speed vs. quality and are important for deployment.

2. Results use gen lengths 256–512; authors may do experiement on L=1-4k. The rollout cost and cache efficacy under long prompts should be evaluated

3. How are layer norms, residuals, and MLP activations treated when some tokens are stale, as freezing only K V may mismatch with new hidden states.

**Questions:**

see weaknesses

---

> ### Author Response · Authors · 2025-11-20
> **Response  to Reviewer KmGv   (Part 1/1)**
>
> # Dear Reviewer KmGv,
> We sincerely appreciate your constructive feedback and valuable suggestions. We include extensive experiments and thorough discussions to address each of your concerns. Our detailed responses are as follows.
>
> ---
>
> ## R2W1: Hyperparameter Analysis
> We sincerely apologize for the confusion caused. To clarify, in the previous submission, we have already conducted hyperparameter analysis on $k$ and $p$, which can be found in Appendix D.4. Furthermore, following Reviewer 4Cqp's suggestion, we perform hyperparameter analysis on Dream-Inst and the HumanEval dataset under both shorter and longer generation lengths (i.e., 256 and 1024). As shown in the table, configurations with $k = 24, 32$, $p = 0.1, 0.2$, and $\sigma = 10$ generally achieve a favorable trade-off between inference throughput and generation quality.
>
> ### **$k$ and $p$ Ablation on Dream-Inst, Humaneval, Gen Len = 256**
>
> | $p \setminus k$ | Metric | $k=16$ | $k=24$ | $k=32$ | $k=40$ |
> | :--- | :--- | :--- | :--- | :--- | :--- |
> | **$p=0.05$** | Score | 59.75 | 60.36 | 60.97 | 58.54 |
> | | Speed | 15.02 | 14.83 | 14.6 | 14.52 |
> | **$p=0.1$** | Score | 60.36 | 58.54 | 64.02 | 57.93 |
> | | Speed | 14.64 | 14.4 | 14.34 | 14.03 |
> | **$p=0.2$** | Score | 59.15 | 57.93 | 59.76 | 58.54 |
> | | Speed | 13.85 | 13.59 | 13.47 | 13.38 |
> | **$p=0.4$** | Score | 62.8 | 59.76 | 59.76 | 57.32 |
> | | Speed | 11.6 | 11.33 | 11.11 | 11 |
>
> ### **$k$ and $p$ Ablation on Dream-Inst, Humaneval, Gen Len = 1024**
> | $p \setminus k$ | Metric | $k=16$ | $k=24$ | $k=32$ | $k=40$ |
> | :--- | :--- | :--- | :--- | :--- | :--- |
> | **$p=0.05$** | Score | 60.36 | 61.56 | 58.54 | 60.37 |
> | | Speed | 13.46 | 13.14 | 12.98 | 12.68 |
> | **$p=0.1$** | Score | 64.64 | 62.8 | 64.02 | 60.36 |
> | | Speed | 12.56 | 12.3 | 11.68 | 11.18 |
> | **$p=0.2$** | Score | 60.36 | 60.98 | 58.54 | 59.15 |
> | | Speed | 9.95 | 9.78 | 9.6 | 9.5 |
> | **$p=0.4$** | Score | 61.58 | 59.77 | 57.93 | 59.76 |
> | | Speed | 7.03 | 6.88 | 7 | 6.89 |
>
> ### **$\sigma$ Ablation on Dream-Inst, Humaneval**
> | GenLen \setminus \sigma$ | $\sigma=1$ | $\sigma=10$ | $\sigma=20$ | $\sigma=40$ | $\sigma=80$ |
> | :--- | :--- | :--- | :--- | :--- | :--- |
> | **Gen Len=256** | **62.2** | 61.6 | 58.54 | 57.93 | 54.88 |
> | **Gen Len=512** | 50 | **61.58** | 59.75 | 57.92 | 50 |
> | **Gen Len=1024** | 52.44 | **61.56** | 57.93 | 58.54 | 60.36 |
>
> ---
>
> ## R2W2: d2Cache Performance on Long Sequences
> Following the reviewer's suggestion, we further evaluate our method on Dream-Inst under (1) longer generation lengths (i.e., 1024 tokens) and (2) longer prompts (i.e., 1.4k  tokens).
> 1. Longer generation length (1024).
> We evaluate the performance of d2Cache on the four datasets used in our main paper.
> 2. Longer prompts (1.4k).
> We evaluate all methods on the MMLU-Pro (5-shot) dataset, where the average prompt length consists of 1.4k tokens.
>
> ### **Long Context Performance on Dream-Inst**
>
> | **Benchmarks** | **Metric** | **Vanilla** | **Fast dLLM** | **dLLM-Cache** | **Ours** |
> | :--- | :--- | :--- | :--- | :--- | :--- |
> | **GSM8k (Gen. Len = 1024)** | Score | 68.46 | 67.85 | **68.76** | 66.29 |
> | | Speed | 1.54 | 4.18 (2.7x) | 3.33 (2.2x)| **8.58 (5.6x)** |
> | | Latency | 671.35 | 245.29 | 308.62 | **119.69** |
> | **Math (Gen. Len = 1024)** | Score | **43.6** | 42.6 | 40.4 | 41.2 |
> | | Speed | 1.89 | 4.43 (2.3x) | 2.75 (1.5x)| **9.55 (5.1x)**  |
> | | Latency | 541.04 | 231.45 | 373.1 | **107.29** |
> | **HumanEval (Gen. Len = 1024)** | Score | 56.71 | 58.53 | 60.97 | **64.02** |
> | | Speed | 2.62 | 4.77 (1.8x) | 3.03 (1.2x)| **11.74 (4.5x)**  |
> | | Latency | 393.47 | 214.5 | 338.21 | **87.64** |
> | **MBPP (Gen. Len = 1024)** | Score | 52.8 | 52.4 | 54.2 | **56.4** |
> | | Speed | 1.95 | 4.45 (2.3x)| 4.59 (2.4x)| **9.76 (5.0x)**  |
> | | Latency | 526.86 | 229.91 | 223.05 | **105.05** |
> | **MMLU (Gen. Len = 256)** | Score | **47.92** | 45.92 | 46.45 | 46.79 |
> | | Speed | 2.15 | 9.04 (4.2x) | 7.12 (3.3x) | **10.12 (4.7x)**  |
> | | Latency | 126.31 | 27.69 | 34.09 | **25.77** |
>
> Across both settings, we observe that other methods—due to their coarse-grained nature—experience severely degraded acceleration as the context length increases. In contrast, d2Cache maintains substantial speedups without performance loss even in long-context scenarios, owing to its fine-grained two-stage token selection. These results clearly demonstrate the generality of d2Cache.
>
> ---
>
> ## R2W3: Potential Hidden-State Mismatching
> We sincerely apologize for the confusion caused. To clarify, after the two-stage token selections of d2Cache, we obtain the token set $y^\ast_{t+1}$ (Eq. 5). Only tokens in $y^\ast_{t+1}$ pass through layer norms, residual connections, and MLP activations, ensuring that their hidden states remain up to date. The remaining stale tokens contribute only through their key–value states, which interact with the representations of $y^\ast_{t+1}$ via self-attention. Therefore, no hidden-state mismatching occurs.

---

> ### Comment · Reviewer_KmGv · 2025-11-21
> **Thank you**
>
> Thank you for addressing my concerns. I will keep my positive score.

---

> > ### Author Response · Authors · 2025-11-22
> >
> > We would like to thank the reviewer for keeping your positive score. We are especially grateful for your recognition of the value of dLLMs and the necessity of advancing efficient dLLM inference. Your positive perspective on this emerging direction is highly encouraging for us and, we believe, for the community as well.
> >
> > From your latest comment, it appears that all the concerns have been fully addressed. If there is anything still unclear or if additional discussions and results could help you further evaluate our work, we would be more than happy to provide them.

---

### Official Review · Reviewer_4Cqp · 2025-10-31

**Soundness:** 3
**Presentation:** 4
**Contribution:** 3
**Rating:** 6
**Confidence:** 3

**Summary:**

Diffusion-based large language models (dLLMs) outperform autoregressive models (ARMs) in mitigating the "reversal curse" and capturing global semantics but suffer from low inference efficiency—they cannot use standard KV caches due to bidirectional attention, and existing coarse-grained approximate KV solutions lack flexibility or cause redundancy. To address this, the authors propose D2Cache, a training-free framework built on two insights: masked tokens’ KV states only need updates in the "rapid-change phase" before decoding, and dLLMs’ attention concentrates on a small subset of prompt/decoded tokens. D2Cache uses a two-stage strategy: 1) selecting masked tokens via a "certainty prior" (combining prediction confidence and nearby known token density) for KV updates, enabling quasi left-to-right generation; 2) choosing high-impact prompt/decoded tokens via attention rollout. Experiments on LLaDA-8B and Dream-v0-7B show D2Cache achieves 3.2×–4.0× average speedups (e.g., 4.7× on Dream-Inst/GSM8K) while maintaining/improving generation quality, outperforming baselines like dLLM-Cache and Fast-dLLM. It advances dLLMs’ practical deployment by bridging performance and inference efficiency.

**Strengths:**

- **Originality**: Breaks from prior coarse-grained (segment/block-level) KV caching for dLLMs, proposing a token-level adaptive strategy. It’s driven by two dLLM-specific insights (masked tokens’ KV states only need updates in the "rapid-change phase"; dLLMs’ attention concentrates on prompt/decoded tokens) and introduces "certainty prior-guided decoding" to fix NAR’s premature overconfidence bias.

- **Quality**: Rigorous validation covers 2 dLLMs (LLaDA-8B, Dream-v0-7B), 4 tasks (reasoning, code generation), and key metrics (throughput, latency, accuracy). Comprehensive ablations (decoding schemes, update phases, hyperparameters) isolate design impacts, with mathematical formalization (Gaussian-weighted density, attention rollout) ensuring theoretical soundness.

- **Clarity**: Follows a logical "problem→insight→solution→validation" flow. Visualizations and plain-language formula explanations enhance accessibility, while full parameter specs, public datasets, and planned code release support reproducibility.

- **Significance**: Achieves 3.2×–4.0× average inference speedups (up to 4.7×) while maintaining accuracy, resolving dLLMs’ deployment bottleneck. The token-level caching paradigm inspires optimization for other bidirectional models, and its benchmarks set a reference for future work.

**Weaknesses:**

+ **Narrow validation scope**: Only tests mid-sized dLLMs (8B/7B) and 4 reasoning/code tasks—lacks large-scale dLLMs (17B+/70B) and non-technical tasks (dialogue/creative writing). If it is possible, add experiments on large dLLMs and conversational datasets (e.g., DailyDialog).

+ **Fixed hyperparameters**: Uses static k=32/p=0.1/σ=10 across all scenarios, no adaptive tuning. If it is possible, add hyperparameter sensitivity analysis (e.g., long vs. short sequences) and lightweight dynamic adjustment (as a supplement to ablation study, beyond section D.4).

+ **Unreported memory overhead**: Ignores peak GPU memory usage—critical for edge deployment. If it is possible, add memory metrics and analyze speed-accuracy-memory trade-offs.

**Questions:**

+ **Discussion 1**: How do you think your method is compatible with token sparsification techniques [1], which have been proven useful for LLMs [2]? Could you add related discussion?

+ **Discussion 2**: While the paper presents the method as a training-free framework, it raises the question of its potential trainability. It would be interesting to investigate whether the framework can be made trainable and how fine-tuning might affect its performance.

**I would be happy to increase my rating if my views are given a thorough discussion.**

[1]: Wu X, Zeng F, Wang X, et al. PPT: Token pruning and pooling for efficient vision transformers[J]. arXiv preprint arXiv:2310.01812, 2023.

[2]: Wan Z, Wu X, Zhang Y, et al. D2o: Dynamic discriminative operations for efficient generative inference of large language models[J]. arXiv preprint arXiv:2406.13035, 2024.

---

> ### Author Response · Authors · 2025-11-20
> **Response to Reviewer 4Cqp   (Part 1/2)**
>
> # Dear Reviewer 4Cqp,
> We sincerely appreciate your constructive feedback and valuable suggestions. We include extensive experiments and thorough discussions to address each of your concerns. Our detailed responses are as follows.
>
> ---
>
> ## R1W1: Larger-Scale Models and General-Purpose Datasets
> **Larger-scale models.** We would like to clarify that diffusion-based large language models (dLLMs) are still an emerging research direction, and only this year such models are successfully scaled to the 7–8B range. For example, LLaDA [1] appeared in February of this year, and Dream [2] was released in May. A more complete timeline can be found in Figure 1 of the recent survey [3]. As a result, the open-source community currently lacks the larger dense models as you mentioned (e.g., 17B+/70B). This is also why all existing dLLM caching works (e.g., Fast-dLLM and dLLM-Cache) focus on LLaDA-8B and Dream-7B. We fully agree that evaluations on larger models would be valuable, and we are open to including such models once they become available.
>
> **General-purpose datasets.** Following the reviewer's suggestion, we evaluate both the baselines and our proposed d2cache on two general-purpose tasks, MMLU and GPQA. These datasets are also used by other baselines, and their hyperparameters are tuned accordingly. As shown in the following table, our method consistently outperforms the baselines in speed on both datasets and achieves performance comparable to vanilla decoding.
>
> ### **Performance of General Tasks on Dream-Inst**
> | Benchmarks | Metric | Vanilla | Fast-dLLM | dLLM-Cache | Ours |
> | :--- | :--- | :--- | :--- | :--- | :--- |
> | GPQA  | Perf | 30.13 | **34.59** | 31.02 | 31.47 |
> | | Speed | 6.43 | 11.75 (1.8x) | 10.91 (1.7x) | **14.65 (2.3x)**  |
> | MMLU  | Perf | **47.92** | 45.92 | 46.45 | 46.79 |
> | | Speed | 2.15 | 9.04 (4.2x) | 7.12 (3.3x) | **10.12 (4.7x)**  |
>
> ---
>
> ## R1W2: Hyperparameter Sensitivity Under Different Sequence Lengths
> Following the reviewer's suggestion, we conduct a grid search with generation lengths of 256 and 1024 on the HumanEval dataset, using $p = 0.05, 0.1, 0.2, 0.4$, $k = 16, 24, 32, 40$ and $\sigma= 1, 10, 20, 40, 80$ with Dream-Inst.
>  In terms of speed, the experimental results clearly show that shorter sequences and smaller $k$  and $p$ lead to higher inference throughput, which aligns with the intuition. In terms of performance, we observe that  $\sigma=10$ generally yields the best results across different tasks and generation lengths. In contrast, we do not observe a clear or consistent pattern for the hyperparameters $p$ and $k$; their optimal values appear to vary across tasks and even across different instances within the same task.This suggests that exploring a lightweight dynamic adjustment mechanism for $k$ and $p$ would be highly beneficial. We further discuss this possibility in our response to Q2.
>
> ### **$k$ and $p$ Ablation on Dream-Inst, Humaneval, Gen Len = 256**
> | $p$\\\\$k$ | Metric | $k=16$ | $k=24$ | $k=32$ | $k=40$ |
> | :--- | :--- | :--- | :--- | :--- | :--- |
> | **$p=0.05$** | Score| 59.75 | 60.36 | 60.97 | 58.54 |
> | | Speed | 15.02 | 14.83 | 14.6 | 14.52 |
> | **$p=0.1$** | Score| 60.36 | 58.54 | 64.02 | 57.93 |
> | | Speed | 14.64 | 14.4 | 14.34 | 14.03 |
> | **$p=0.2$** | Score| 59.15 | 57.93 | 59.76 | 58.54 |
> | | Speed | 13.85 | 13.59 | 13.47 | 13.38 |
> | **$p=0.4$** | Score| 62.8 | 59.76 | 59.76 | 57.32 |
> | | Speed | 11.6 | 11.33 | 11.11 | 11 |
>
> ### **$k$ and $p$ Ablation on Dream-Inst, Humaneval, Gen Len=1024**
>
> | $p$\\\\$k$ | Metric | $k=16$ | $k=24$ | $k=32$ | $k=40$ |
> | :--- | :--- | :--- | :--- | :--- | :--- |
> | **$p=0.05$** | Score| 60.36 | 61.56 | 58.54 | 60.37 |
> | | Speed | 13.46 | 13.14 | 12.98 | 12.68 |
> | **$p=0.1$** | Score| 64.64 | 62.8 | 64.02 | 60.36 |
> | | Speed | 12.56 | 12.3 | 11.68 | 11.18 |
> | **$p=0.2$** | Score| 60.36 | 60.98 | 58.54 | 59.15 |
> | | Speed | 9.95 | 9.78 | 9.6 | 9.5 |
> | **$p=0.4$** | Score| 61.58 | 59.77 | 57.93 | 59.76 |
> | | Speed | 7.03 | 6.88 | 7 | 6.89 |
>
> ### **$\sigma$ Ablation on Dream-Inst, Humaneval**
>
> | GenLen\\\\$\sigma$ | $\sigma=1$ | $\sigma=10$ | $\sigma=20$ | $\sigma=40$ | $\sigma=80$ |
> | :--- | :--- | :--- | :--- | :--- | :--- |
> | **Gen Len=256** | **62.2** | 61.6 | 58.54 | 57.93 | 54.88 |
> | **Gen Len=512** | 50 | **61.58** | 59.75 | 57.92 | 50 |
> | **Gen Len=1024** | 52.44 | **61.56** | 57.93 | 58.54 | 60.36 |

---

> ### Author Response · Authors · 2025-11-20
> **Response to Reviewer 4Cqp (Part 2/2)**
>
> ## R1W3: Peak  Memory Usage of d2Cache
> **Memory consumption.** Following the reviewer’s suggestion, we conduct a thorough analysis and profiling of the peak memory usage of d2cache. Note that the KV cache used by dLLMs consumes the same amount of memory as that required by an autoregressive LLM (ARM) of the same scale. Specifically, for sequence length $L$, number of layers $N$, and hidden dimension $d$, an ARM or a dLLM stores $2 \times L \times N \times d$ floating-point values for the KV cache. d2Cache additionally stores an attention-rollout matrix of size $L \times L$
> which is typically negligible. We report the peak memory usage on Dream Inst for a generation length of 1024 across four datasets. As shown in the table, for example on GSM8K, where the average prompt length is approximately 800—resulting in a sequence length of roughly 1.8k—the additional memory consumption of d2Cache is nearly identical to that of Fast dLLM.
> ### **Memory Usage (GB) on Dream-Inst, Gen Len=1024**
>
> | Benchmarks | Vanilla | Fast dLLM | dllm-Cache | Ours |
> | :--- | :--- | :--- | :--- | :--- |
> | GSM8k | 19.26 | 19.39 | 20.28 | 19.35 |
> | Math | 19.15 | 19.27 | 20.2 | 19.28 |
> | HumanEval | 19.06 | 19.16 | 19.79 | 19.14 |
> | MBPP | 19.12 | 19.23 | 19.94 | 19.24 |
>
> **Speed–accuracy–memory trade-offs.** Because the hyperparameters of our method do not affect memory usage, there is no trade-off involving memory. For the trade-off between speed and accuracy, please refer to W2.
>
> ---
>
> ## R1Q1: Compatibility with Token Sparsification Techniques
> As shown in [4], the attention rollout mechanism is simply a way to identify important tokens and can be seamlessly replaced by other token-sparsification techniques.
> Thank you for the suggestion—we will include a discussion of the paper you mentioned in the Related Work section.
>
> ---
>
> ## R1Q2: Integration to a Trainable Framework
> Thank you for the suggestion. We outline several possible ways to make certain components trainable; however, due to time and resource limitations during the rebuttal period, we were unable to experimentally verify them.
> - In Stage 1 of d2cache, our method first selects masked tokens that lie in the rapid-change phase. Therefore, one may train a simple linear layer to classify whether a token is in this phase based on its last hidden state. Once trained, ( k ) would no longer need to be manually specified; instead, the model itself can dynamically identify and update only the masked tokens that genuinely require refinement, thereby improving computational efficiency.
> - In Stage 2 of d2cache, our method further selects pivotal tokens among both prompt and decoded tokens. Similarly, a linear layer can be trained to predict each token’s contribution to generation using its last hidden state, which thus avoids the need to manually specify ( p ). This enables dynamic identification of pivotal tokens at each decoding step.
>
> ---
>
> ## Reference
>
> [1] Nie S, Zhu F, You Z, et al. Large language diffusion models[J]. arXiv preprint arXiv:2502.09992, 2025.
>
> [2] Ye J, Xie Z, Zheng L, et al. Dream 7b: Diffusion large language models[J]. arXiv preprint arXiv:2508.15487, 2025.
>
> [3] Yu R, Li Q, Wang X. Discrete Diffusion in Large Language and Multimodal Models: A Survey[J]. arXiv preprint arXiv:2506.13759, 2025.
>
> [4] Abnar S, Zuidema W. Quantifying attention flow in transformers[J]. arXiv preprint arXiv:2005.00928, 2020.

---

> > ### Comment · Reviewer_4Cqp · 2025-11-24
> > **Response to Rebuttal and Score Update**
> >
> > I apologize for not replying sooner over the past few days, as I have been quite busy. Thank you for your thoughtful and detailed response to my comments. Since you have carefully addressed my concerns, I have raised my score accordingly.

---

> > > ### Author Response · Authors · 2025-11-24
> > >
> > > Thank you very much for taking the time to review our response despite your busy schedule. We sincerely appreciate your careful reading and your recognition of our efforts.

---

> ### Comment · Area_Chair_GPdc · 2025-11-23
>
> Dear reviewer,
>
> Thanks for your time and effort in reviewing ICLR2026 submissions. The authors have submitted their responses to your review. Please take the time to read and raise your further comments, and discuss with the authors.
>
> Best regards,
>
> AC

---

### Author Response · Authors · 2025-11-30
**Summary of Discussions with Reviewers Prior to the Leakage Incident**

Dear Area Chair,

We sincerely appreciate you stepping in to handle our submission. We understand that this requires significant additional effort, and we are truly grateful for your time and support.


Prior to the leakage incident, we had already engaged in active discussions with all four reviewers. Among them, three reviewers explicitly indicated that they maintain positive evaluations/scores, as reflected in their comments:

- Reviewer 4Cqp raised the score from 6 to 8 and stated: “**Since you have carefully addressed my concerns, I have raised my score accordingly.**”

- Reviewer KmGv maintained the score of 6 and stated: “**Thank you for addressing my concerns. I will keep my positive score.**”

- Reviewer 1oge kept positive evaluation and stated: “**I appreciate your response to the questions about d2Cache’s large-model scalability, long-sequence adaptation, and failure case analysis, and I’ll keep my positive evaluation.**”

The remaining reviewer also revised the overall evaluation/score:

- Reviewer E2JM increased the score from 2 to 4 and stated: “**My doubts about the paper's motivation have been solved. Therefore, I will increase my score to 4.**” We have also addressed all additional concerns raised by Reviewer E2JM, which can be found in R4Opinion 1&2 and R4Reason 1&2.

Thank you again for your time and effort.

Best regards,\
Anonymous Authors

---

### Author Response · Authors · 2025-12-03
**Summary of Revisions**

Dear Area Chair,

Thank you very much for your time and effort in managing the review process for the conference. In accordance with the reviewers' comments, we have incorporated extensive experiments and thorough discussions in the revised manuscript to clarify the technical contributions of our work.

For your convenience, below we summarize where and how each comment has been taken into account.

---

**Additional experiments and ablations**

- R1W1, R4W3: Performance on general tasks. (**resolved, see Table 1 and Table 5**)
- R1W2, R2W1, R3Q2: Hyperparameter sensitivity under different sequence lengths. (**resolved, see Section 5.3 and Figure 6**)
- R1W3: Peak memory usage comparison. (**resolved, see Appendix D.1 and Table 7**)
- R2W2, R3W1&Q2: Performance on long sequences. (**resolved, see Appendix E.3 and Table 7**)
- R3W2&Q3: Failure case root cause on LLaDA-Inst/MBPP. (**resolved, see Appendix D.3 and Figure 8**)
- R4W2: Results under the same decoding strategy. (**resolved, see Appendix E.2 and Table 6**)

---

**Additional clarifications on our work**

- R1Q1: Compatibility with token sparsification techniques. (**resolved, see Section 2**)
- R4W1&W2&Q2&Option1: Potential unfair comparisons. (**resolved, see Section 5.1**)

---

**Questions and discussions**

- R1W1, R3W1&Q1: Larger scale models. (**resolved, see Appendix D.4**)
- R1Q2, R3W1&Q1, R4Reason2: Integration to a trainable framework or a lightweight variant. (**resolved, see Appendix D.4 and Table 4**)
- R2W3: Potential hidden-state mismatching. (**resolved, see Section 4.2**)
- R4Q1: Difference between quasi-AR and AR systems. (**resolved, see Appendix D.2**)

---

Thank you again for your time and effort.

Best regards,\
Anonymous Authors

---

### Meta-Review · Area_Chair_k1Nj · 2025-12-21

**Summary:**

This paper presents D2Cache, a training-free, fine-grained KV caching framework for accelerating inference in diffusion-based large language models (dLLMs). The work is motivated by the inefficiency of vanilla dLLM decoding and proposes a two-stage token selection mechanism based on certainty priors and attention rollout to selectively refresh KV states.

Across the review process, reviewers broadly agreed that the method is technically sound, well-motivated, and empirically validated. While initial concerns were raised regarding evaluation scope, hyperparameter sensitivity, comparison fairness, and long-context behavior, the authors addressed these points through extensive additional experiments, clarifications, and analyses in the rebuttal. As a result, most reviewers converged toward a positive assessment of the paper’s technical quality and contribution.

**Reviewer Concerns:**

**Concerns addressed by the rebuttal:**

Added evaluations on general-purpose benchmarks and longer contexts, clarifying the method’s applicability beyond short reasoning tasks.

Provided hyperparameter sensitivity analyses and discussed potential adaptive extensions.

Reported peak memory usage and clarified speed–accuracy–memory trade-offs.

Resolved ambiguity in baseline comparisons by evaluating methods under consistent decoding strategies.

Investigated and explained failure cases (e.g., MBPP) with concrete qualitative analysis.

Clarified the relationship between quasi-autoregressive decoding in dLLMs and standard AR inference.

**Remaining concerns:**

One reviewer remained unconvinced about the breadth of impact and viewed the contribution as incremental within the dLLM acceleration line of work.

Questions about long-term influence beyond the dLLM setting were raised, though these concerns were more about scope and positioning rather than correctness.

Overall, the remaining concerns reflect differences in judgment about impact, rather than unresolved technical flaws.

**Reviewer Scores:**

Reviewer 4Cqp: Initially borderline, raised the score after rebuttal and expressed clear satisfaction with the revisions.

Reviewer KmGv: Maintained a positive score after rebuttal, indicating all concerns were addressed.

Reviewer 1oge: Expressed a positive evaluation after rebuttal, though the numerical score was not updated accordingly.

Reviewer E2JM: Raised the score from a strong reject to borderline, but continued to recommend rejection based on perceived community-level impact rather than technical issues.

Thus, three reviewers ultimately expressed positive evaluations, with one remaining negative primarily on subjective impact grounds.

---

### Decision · Program_Chairs · 2026-01-26

Accept (Poster)